# A FUNCTIONAL PERSPECTIVE ON MULTI-LAYER OUT-OF-DISTRIBUTION DETECTION

## ABSTRACT

A crucial component for implementing reliable classifiers is detecting examples far from the reference (training) distribution, referred to as out-of-distribution (OOD) samples. A key feature of OOD detection is to exploit the network by extracting statistical patterns and relationships through the pre-trained multi-layer classifier. Despite achieving solid results, state-of-the-art methods require either additional OOD examples, expensive computation of gradients, or are tightened to a particular architecture, limiting their applications. This work adopts an original approach based on a functional view of the network that exploits the sample's trajectories through the various layers and their statistical dependencies. In this new framework, OOD detection translates into detecting samples whose trajectories differ from the typical behavior characterized by the training set. Our method significantly decreases the OOD detection error of classifiers trained on ImageNet and outperforms the state-of-the-art methods on average AUROC and TNR at 95% TPR. We demonstrate that the functional signature left by a sample in a network carries relevant information for OOD detection.

## 1 INTRODUCTION

The ability of a Deep Neural Network (DNN) to generalize to new data is mainly restricted to priorly known concepts in the training dataset. In real-world scenarios, Machine Learning (ML) models may encounter Out-Of-Distribution (OOD) samples, such as data belonging to novel concepts (classes) (Pimentel et al., 2014), abnormal samples (Tishby & Zaslavsky, 2015), or even carefully crafted attacks designed to exploit the model (Szegedy et al., 2013). The behavior of ML systems on unseen data is of great concern for safety-critical applications (Amodei et al., 2016b;a), such as medical diagnosis in healthcare (Subbaswamy & Saria, 2020), autonomous vehicle control in transportation (Bojarski et al., 2016), among others. To address safety issues arising from the presence of OOD samples, a successful line of work aims at augmenting ML models with an OOD binary detector to distinguish between abnormal and in-distribution examples (Hendrycks & Gimpel, 2017). An analogy to the detector is the human body's immune system, with the task of differentiating between antigens and the body itself.

Distinguishing OOD samples is challenging. Some previous works developed detectors by combining scores at the various layers of the multi-layer pre-trained classifier (Sastry & Oore, 2020; Lee et al., 2018; Gomes et al., 2022; Huang et al., 2021). These detectors require either a held-out OOD dataset (e.g., adversarially generated or OOD data) or ad-hoc methods to linearly combine OOD scores computed on each layer embedding tightened to a particular architecture. A key observation is that existing aggregation techniques overlook the sequential nature of the underlying problem and, thus, limit the discriminative power of those methods. Indeed, an input sample passes consecutively through each layer and generates a highly correlated signature that can be statistically characterized. Our observations in this work motivate the statement that:

*The input's trajectory through a network is key for discriminating typical from atypical samples.*

In this paper, we introduce a significant change of perspective. Instead of looking at each layer score independently, we cast the scores into a sequential representation that captures the statistical trajectory of an input sample through the various layers of a multi-layer neural network. To this end, we adopt a functional point of view by considering the sequential representation as curves parametrized

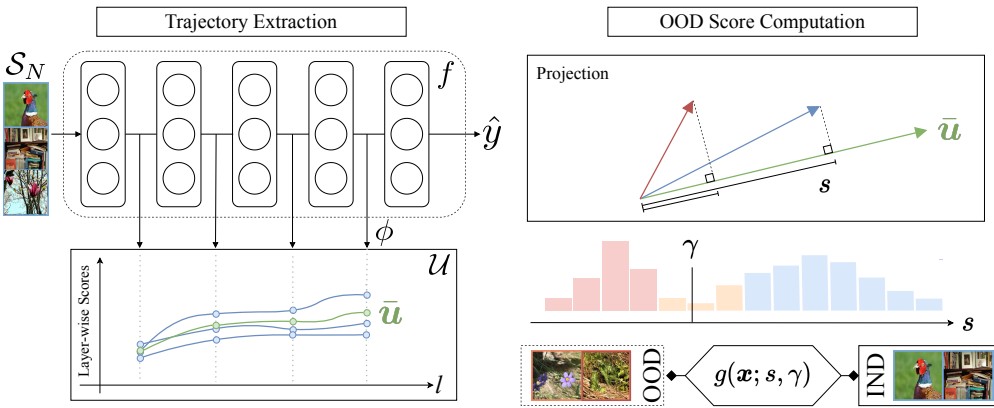

Figure 1: The left-hand side of the figure shows the feature extraction process of a deep neural network classifier $f$. The mapping of the hidden representations of an input sample into a functional representation is given by a function $\phi$. The functional representation of a sample encodes valuable information for OOD detection. The right hand side of the figure shows how our method computes the OOD score $s$ of a sample during test time. A projection of the sample's trajectory into the training reference trajectory $\bar{u}$ is computed. Finally, a threshold $\gamma$ is set to obtain a binary discriminator $g$.

by each layer. Consequently, we redefine OOD detection as detecting samples whose trajectories are abnormal (or atypical) compared to reference trajectories characterized by the training set. Our method, which requires little parameter tuning and, perhaps more importantly, no additional OOD or synthetic data, can identify OOD samples from their trajectories. Furthermore, we show that typical multivariate detection methods fail to detect OOD patterns, which may manifest in an isolated fashion by shifting in magnitude or overall shape. Figure 1 summarizes our method.

**Contributions.** This work presents a new principle and unsupervised method for detecting OOD samples that do not require OOD (or extra) data and brings novel insights into the problem of OOD detection. Our main contributions can be summarized as follows.

1. *A novel problem formulation.* We reformulate the problem of OOD detection through a functional perspective that effectively captures the statistical dependencies of an input sample's path across a multi-layer neural classifier. Moreover, we propose a map from the multivariate feature space (at each layer) to a functional space that relies on the probability weighted projection of the test sample onto the class conditional training prototypes at the layer. It is computationally efficient and straightforward to implement.

2. *Computing OOD scores from trajectories.* We compute the inner product between the test trajectory and the average training trajectories to measure the similarity of the input w.r.t the training set. Low similarity indicates that the test sample is likely sampled from OOD.

3. *Empirical evaluation.* We validate the value of the proposed method using a mid-size OOD detection benchmark on ImageNet-1k. We obtain competitive results, demonstrating an average ROC gain of 3.7% across three architectures and four OOD datasets. We release our code anonymized online.

## 2 RELATED WORKS

This section briefly discusses prior work in OOD detection, highlighting confidence-based and feature-based methods without special training as they resonate the most with our work. Another thread of research relies on re-training for learning representations adapted to OOD detection (Mohseni et al., 2020; Bitterwolf et al.; Mahmood et al., 2021), either through contrastive training (Hendrycks et al., 2019; Winkens et al., 2020; Sehwag et al., 2021), regularization (Lee et al., 2021; Nandy et al., 2021; Hein et al., 2019; Du et al., 2022), generative (Schlegl et al., 2017; ood, 2019; Xiao et al.; Ren et al.; Zhang et al., 2021), or ensemble (Vyas et al., 2018; Choi & Jang, 2018) based approaches. Related subfields are open set recognition (Geng et al., 2021), novelty detection (Pi-

mentel et al., 2014), anomaly detection (Chandola et al., 2009; Chalapathy & Chawla, 2019), outlier detection (Hodge & Austin, 2004), and adversarial attacks detection (Akhtar & Mian, 2018).

**Confidence-based OOD detection.** A natural measure of uncertainty of a sample's label is the classification model's softmax output. Hendrycks & Gimpel (2017) observed that the maximum of the softmax output could be used as a discriminative score between in-distribution and OOD samples. Hein et al. (2019) observed that it may still assign overconfident values to OOD examples. Liang et al. (2018) and Hsu et al. (2020) propose re-scaling the softmax response with a temperature value and a pre-processing technique that further separates in- from out-of-distribution examples. Liu et al. (2020) proposes an energy-based OOD detection score by replacing the softmax confidence score with the free energy function. Sun & Li (2022) proposes sparsification of the classification layer weights to improve OOD detection by regularizing predictions.

**Feature-based OOD detection.** This line of research focuses on exploring latent representations for OOD detection. For instance, Haroush et al. (2021) considers using statistical tests, Sastry & Oore (2020) rely on higher-order Grams matrices, Quintanilha et al. (2019) uses mean and standard deviation within feature maps, Sun et al. (2021) proposes clipping the activations to boost OOD detection performance, while recent work Huang et al. (2021) also explores the gradient space. Normalizing and residual flows to estimate the probability distribution of the feature space were proposed in Kirichenko et al. and Zisselman & Tamar (2020). Lin et al. (2021) rely on networks with multiple classifiers and introduces a dynamic OOD detection method which selects the index of the layer in test time to compute the score from. Sun et al. (2022) proposes non-parametric nearest-neighbor based on the Euclidean distance for OOD detection. Perhaps one of the most widely used techniques relies on the Gaussian mixture assumption for the hidden representations and the Mahalanobis distance (Lee et al., 2018; Ren et al., 2021) or further information geometry tools (Gomes et al., 2022). Efforts toward combining multiple features to improve performance were previously explored (Lee et al., 2018; Sastry & Oore, 2020; Gomes et al., 2022). The strategy relies heavily upon having additional data for tuning the detector or focusing on specific model architectures, which are limiting factors in real-world applications.

## 3 PRELIMINARIES

We start by recalling the general setting of the OOD detection problem from a mathematical point of view (Section 3.1). Then, in Section 3.2, we motivate our method through a simple yet clarifying illustrative example showcasing the limitation of previous works and how we approach the problem.

### 3.1 BACKGROUND

Let $(X, Y)$ be a random variable valued in a space $\mathcal{X} \times \mathcal{Y}$ with unknown probability density function (pdf) $p_{XY}$ and probability distribution $P_{XY}$. Here, $\mathcal{X} \subseteq \mathbb{R}^d$ represents the covariate space and $\mathcal{Y} = \{1, \ldots, C\}$ corresponds to the labels attached to elements from $\mathcal{X}$. The training dataset $\mathcal{S}_N = \{(\boldsymbol{x}_i, y_i)\}_{i=1}^N$ is defined as independent and identically distributed (i.i.d) realizations of $P_{XY}$. From this formulation, detecting OOD samples boils down to building a binary rule $g : \mathcal{X} \to \{0, 1\}$ through a soft scoring function $s : \mathcal{X} \to \mathbb{R}$ and a threshold $\gamma \in \mathbb{R}$. Namely, a new observation $\boldsymbol{x} \in \mathcal{X}$ is then considered as *in-distribution*, i.e., generated by $P_{XY}$, when $g(\boldsymbol{x}) = 0$ and as OOD when $g(\boldsymbol{x}) = 1$. Finding this rule $g$ from $\mathcal{X}$ can become intractable when the dimension $d$ is large. Thus, previous work rely on a multi-layer pre-trained classifier $f_\theta : \mathcal{X} \to \mathcal{Y}$ defined as:

$$f_\theta(\cdot) = h \circ f_L \circ f_{L-1} \circ \cdots \circ f_1(\cdot),$$

with $L \geq 1$ layers, where $f_\ell : \mathbb{R}^{d_{\ell-1}} \to \mathbb{R}^{d_\ell}$ is the $\ell$-th layer of the multi-layer neural classifier, $d_\ell$ denotes the dimension of the latent space induced by the $\ell$-th layer ($d_0 = d$), and $h$ indicates the classifier that outputs the logits. We also define $\boldsymbol{z}_\ell = (f_\ell \circ \cdots \circ f_1)(\boldsymbol{x})$ as the latent vectorial representation at the $\ell$-th layer for an input sample $\boldsymbol{x}$. We will refer to the logits as $\boldsymbol{z}_{L+1}$ and $h$ as $f_{L+1}$ to homogenize notation. It is worth emphasizing that the trajectory of $(\boldsymbol{z}_1, \boldsymbol{z}_2, \ldots, \boldsymbol{z}_{L+1})$ corresponding to a test input $\boldsymbol{x}_0$ are dependent random variables whose joint distribution strongly depends on the underlying distribution of the input.

Therefore, the design of function $g(\cdot)$ is typically based on the three key steps:

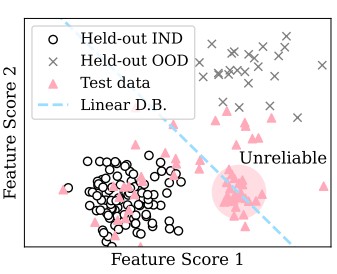
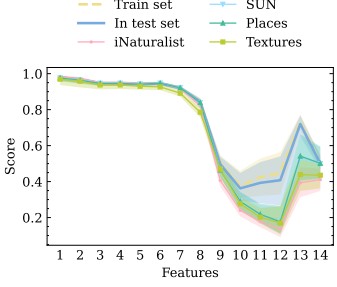
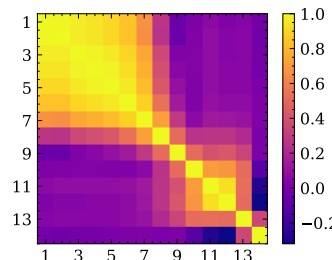

(a) Example of a mispecified model in a toy example in 2D caused by fitting with held-out OOD dataset.

(b) Trajectory of data through a network with 25% and 75% percentile bounds.

(c) Correlation between layers training scores in a network, highlighting structure in the trajectories.

Figure 2: Figure 2a summarizes the limitation of supervised methods for aggregating layer scores that rely on held-out OOD or pseudo-OOD data. It biases the decision boundary (D.B) that doesn't generalize well to other types of OOD data. We observed that in-distribution and OOD data have disparate trajectories through a network (Fig. 2b), specially on the last five features. These features are correlated in a sequential fashion, as observed in Fig. 2c.

(i) A similarity measure $d(\cdot\,;\cdot)$ (e.g., Cosine similarity, Mahalanobis distance, etc.) between a sample and a population is applied at each layer to measure the similarity (or dissimilarity) of a test input $x_0$ at the $\ell$-th layer $z_{\ell,0} = (f_\ell \circ \cdots \circ f_1)(x_0)$ w.r.t. the population of the training examples observed at the same layer $\{z_\ell = (f_\ell \circ \cdots \circ f_1)(x) : x \in \mathcal{S}_N\}$.

(ii) The layer-wise score obtained is mapped to the real line collecting the OOD scores.

(iii) Lastly, a threshold is chosen to build the final decision function.

A fundamental ingredient remains in step (ii):

*How to consistently leverage the information collected from multiple layers outputs in an unsupervised way, i.e., without resorting to OOD or pseudo-OOD examples?*

## 3.2 FROM INDEPENDENT MULTI-LAYER SCORES TO A SEQUENTIAL PERSPECTIVE OF OOD DETECTION

Previous multi-feature OOD detection works treat step (ii) as a supervised learning problem (Lee et al., 2018; Gomes et al., 2022) for which the solution is a linear binary classifier. The objective is to find a linear combination of the scores obtained at each layer that will sufficiently separate in-distribution from OOD samples. A held-out OOD dataset is collected from true (or pseudo-generated) OOD samples. The linear soft novelty score function $s_\alpha$ writes:

$$s_\alpha(x_0) = \sum_{\ell=1}^{L} \alpha_\ell \cdot d\left(x_0; \{(f_\ell \circ \cdots \circ f_1)(x) : x \in \mathcal{S}_N\}\right).$$

The shortcomings of this method are the need for extra data or ad-hoc parameters, which results in decision boundaries that underfit the problem and fail to capture certain types of OOD samples. To illustrate this phenomenon, we designed a toy example (see Figure 2a) where scores are extracted from two features fitting a linear discriminator on held-out in-distribution (IND) and OOD samples. As a consequence, areas of unreliable predictions where OOD samples cannot be detected due to the misspecification of the linear model arise. One could simply introduce a non-linear discriminator that better captures the geometry of the data for this 2D toy example. However, it becomes challenging as we move to higher dimensions with limited data.

By reformulating the problem from a functional data point of view, we can identify trends and typicality in trajectories extracted by the network from the input. Figure 2b shows the dispersion of trajectories coming from the in-distribution and OOD samples. These patterns are extracted from multiple latent representations and aligned on a time-series-like object. *We observed that trajectories coming from OOD samples exhibit a different shape when compared to typical trajectories from*

*training data*. Thus, to determine if an instance belongs to in-distribution, we can test if the observed path is similar to the functional trajectory reference extracted from the training set.

## 4 FUNCTIONAL OUT-OF-DISTRIBUTION DETECTION

This section presents our OOD detection framework, which applies to any pre-trained multi-layer neural network with no requirements for OOD samples. We describe our method through two key steps: functional representation of the input sample (see Section 4.1) and test time OOD score computation (see Section 4.2).

### 4.1 TOWARDS A FUNCTIONAL REPRESENTATION

The first step to obtain an univariate functional representation of the data from the multivariate hidden representations is to reduce each feature map to a scalar value. To do so, we first compute the class-conditional training population prototypes defined by:

$$\boldsymbol{\mu}_{\ell,y} = \frac{1}{N_y} \sum_{i=1}^{N_y} \boldsymbol{z}_{\ell,i}, \tag{1}$$

where $N_y = \big|\{\boldsymbol{z}_{\ell,i} : y_i = y, \forall i \in \{1..N\}\}\big|$, $1 \leq \ell \leq L+1$ and $\boldsymbol{z}_{\ell,i} = (f_\ell \circ \cdots \circ f_1)(\boldsymbol{x}_i)$.

Given an input example, we compute the probability weighted scalar projection[1] between its features (including the logits) and the training class conditional prototypes, resulting in $L+1$ scalar scores:

$$\mathrm{d}_\ell(\boldsymbol{x}; \mathcal{M}_\ell) = \sum_{y=1}^{C} \sigma_y(\boldsymbol{x}) \cdot \mathrm{proj}_{\boldsymbol{\mu}_{\ell,y}} \boldsymbol{z}_\ell = \sum_{y=1}^{C} \sigma_y(\boldsymbol{x}) \|\boldsymbol{z}_\ell\| \cos\big(\angle\,(\boldsymbol{z}_\ell, \boldsymbol{\mu}_{\ell,y})\big), \tag{2}$$

where $\mathcal{M}_\ell = \{\boldsymbol{\mu}_{\ell,y} : y \in \mathcal{Y}\}$, $\|\cdot\|$ is the $\ell_2$-norm, $\angle\,(\cdot, \cdot)$ is the angle between two vectors, and $\sigma_y(\boldsymbol{x}; f_\theta)$ is the softmax function on the logits $f_\theta(\boldsymbol{x})$ of class $y$. Hence, our layer-wise scores rely on the notions of vector length and angle between vectors, which can be generalized to any $n$-dimensional inner product space without imposing any geometrical constraints.

It is worth emphasizing that our layer score has some advantages compared to the class conditional Gaussian model first introduced in Lee et al. (2018) and the Gram matrix based-method introduced in Sastry & Oore (2020). Our layer score encompasses a broader class of distributions as we do not suppose an specific underlying probability distribution. We avoid computing covariance matrices, which are often ill-conditioned for latent representations of DNNs. Since we do not store covariance matrices, our functional approach has a negligible overhead regarding memory requirements. Also, our method can be applied to any vector-based hidden representation, not being restricted to matrix-based representations as in Sastry & Oore (2020). Thus, our approach applies to a broader range of models, including transformers.

By computing the scalar projection at each layer, we define a following *functional neural-representation* extraction function given by Eq. 3. Thus, we can map samples from the representation to a functional space while retaining information on the typicality of the sample w.r.t the training dataset.

$$\begin{aligned} \phi : \mathcal{X} &\to \mathbb{R}^{L+1} \\ \boldsymbol{x} &\mapsto \big[\mathrm{d}_1\,(\boldsymbol{x}; \mathcal{M}_1), \dots, \mathrm{d}_{L+1}\,(\boldsymbol{x}; \mathcal{M}_{L+1})\big] \end{aligned} \tag{3}$$

We apply $\phi$ to the training input $\boldsymbol{x}_i$ to obtain the representation of the training sample across the network $\boldsymbol{u}_i = \phi(\boldsymbol{x}_i)$. We consider the related vectors $\boldsymbol{u}_i, \forall\ i \in [1 : N]$ as curves parameterized by the layers of the network. We build a training dataset $\mathcal{U} = \{\boldsymbol{u}_i\}_{i=1}^{N}$ from these functional representations that will be useful for detecting OOD samples during test time. We then rescale the training set trajectories w.r.t the maximum value found at each coordinate to obtain layer-wise scores on the same scaling for each coordinate. Hence, for $j \in \{1, \dots, L+1\}$, let $\max(\mathcal{U}) := [\max_i u_{i,1}, \dots, \max_i u_{i,L+1}]^\top$, we can compute a reference trajectory $\bar{\boldsymbol{u}}$ for the entire

---

[1]Other metrics to measure the similarity of an input w.r.t. the population of examples can also be used.

training dataset defined in equation 4 that will serve as a global *typical reference* to compare test trajectories with.

$$\bar{\boldsymbol{u}} = \frac{1}{N} \sum_{i=1}^{N} \frac{\boldsymbol{u}_i}{\max(\mathcal{U})} \tag{4}$$

## 4.2 COMPUTING THE OOD SCORE AT TEST TIME

At inference time, we first re-scale the test sample's trajectory as we did with the training reference $\tilde{\phi}(\boldsymbol{x}) = \phi(\boldsymbol{x})/\max(\mathcal{U})$. Then, we compute a similarity score w.r.t this typical reference that will be our OOD score. We choose as metric also the scalar projection of the test vector to the training reference. In practical terms, it boils down to the *inner product* between the test sample's trajectory and the training set's typical reference trajectory since the norm of the average trajectory is constant for all test samples. Mathematically, our scoring function $s : \mathcal{X} \mapsto \mathbb{R}$ writes:

$$s(\boldsymbol{x}; \bar{\boldsymbol{u}}) = \langle \tilde{\phi}(\boldsymbol{x}), \bar{\boldsymbol{u}} \rangle = \sum_{j=1}^{L+1} \tilde{\phi}(\boldsymbol{x})_j \bar{\boldsymbol{u}}_j \tag{5}$$

which is bounded by Cauchy-Schwartz's inequality. From this OOD score, we can derive a binary classifier $g$ by fixing a threshold $\gamma \in \mathbb{R}$:

$$g(\boldsymbol{x}; s, \gamma) = \left\{ \begin{array}{ll} 1, & \text{if } s(\boldsymbol{x}) \leq \gamma \\ 0, & \text{otherwise,} \end{array} \right. \tag{6}$$

where $g(\boldsymbol{x}) = 1$ means that the input sample $\boldsymbol{x}$ is classified as being out-of-distribution. Please refer to Appendix (see Section A.1) for further details on the algorithm.

## 5 EXPERIMENTAL SETTING

This section describes the experimental setting, including the datasets used, the pre-trained DNN architectures, the evaluation metrics, and other essential information. We open-source our code in https://github.com/ood-trajectory/ood-trajectory.

### 5.1 DATASETS

We set as *in-distribution* dataset *ImageNet-1k* (= ILSVRC2012; Deng et al., 2009) for all of our experiments, which is a challenging mid-size and realistic dataset that have been incorporated recently to OOD detection. It contains around 1.28M training samples and 50,000 test samples belonging to 1000 different classes. For the *out-of-distribution* datasets, we take the same dataset splits introduced by Huang & Li (2021). The *iNaturalist* (Horn et al., 2017) dataset contains over 5,000 species of plant and animals. We consider a split with 10,000 test samples with concepts from 110 classes different from the in-distribution ones. The *Sun* (Xiao et al., 2010) dataset is a scene dataset of 397 categories. We considered a split with 10,000 randomly sampled test examples belonging to 50 categories. The *Places365* (Zhou et al., 2017) is also a scenes dataset with 365 different concepts. We also considered a random split with 10,000 samples from 50 disjoint categories. For the DTD or *Textures* (Cimpoi et al., 2014) dataset is composed of textural patterns. We considered all of the 5,640 available test samples. Note that there is a few overlaps between the semantics of classes from this dataset and ImageNet. We decided to keep the entire dataset in order to be comparable with Huang et al. (2021). We provide a study case on this in Section 6.

### 5.2 MODELS

We ran experiments with three models of different architectures. A *DenseNet-121* (Huang et al., 2017) pre-trained on ILSVRC-2012 with 8M parameters and test set top-1 accuracy of 74.43%. We reduced the intermediate representations from the transition blocks with an *max pooling* operation obtaining a final vector with a dimension equal to the number of channels of each output. The dimensions are 128, 256, 512, and 1024, respectively. We resize input images to $224{\times}224$ pixels. A *BiT-S-101* (Kolesnikov et al., 2020) model based on a ResNetv2-101 architecture with top-1 test set

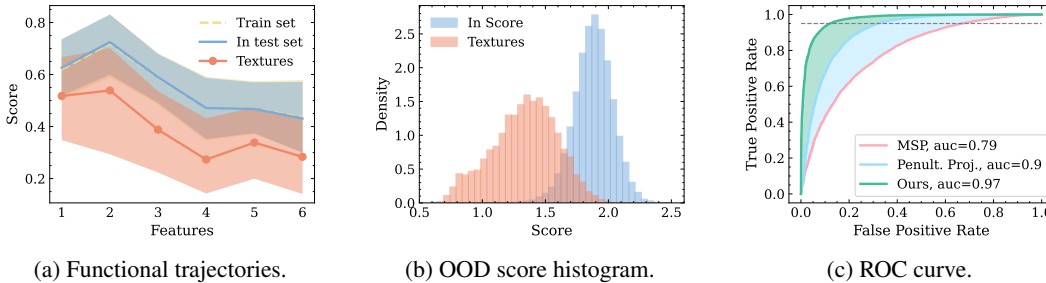

| (a) Functional trajectories. | (b) OOD score histogram. | (c) ROC curve. |

Figure 3: Functional representation with 5 and 95% quantiles, histogram and ROC curve for our OOD score on a DenseNet-121 model with Textures as OOD dataset.

accuracy of 77.41% and 44.5M parameters. We extract features from the outputs of layers 1 to 4 and the penultimate layer, obtaining representations with sizes 256, 512, 1024, and 2048, respectively after max pooling. We resize input images to 480×480. We also ran experiments with a Vision Transformer (*ViT-B-16*; Dosovitskiy et al., 2021), which is trained on the ILSVRC2012 dataset with 82.64% top-1 test accuracy and 70M parameters. We take the output's class tokens for layers 1 to 13 and the encoder's output as latent representations, totaling 14 features of dimension 768 each. We resize images to 224×224. We download the DenseNet and Vision Transformer weights from PyTorch (Paszke et al., 2019) hub and the weights of Big Transfer from Kolesnikov et al. (2020). All models are trained from scratch on ImageNet-1k.

### 5.3 EVALUATION METRICS

We evaluate the methods in terms of ROC and TNR. The Area Under The Receiving Operation Curve (ROC) is the Area Under the curve representing the true negative rate against the false positive rate when considering a varying threshold. It measures how well can the OOD score distinguish between out- and in-distribution data in a threshold-independent manner. The True Negative Rate at 95% True Positive Rate (TNR at 95% TPR or TNR for short) is a threshold-dependent metric that provides the detector's performance in a reasonable choice of threshold. It measures the accuracy in detecting OOD samples when the accuracy of detecting in-distribution samples is fixed at 95%. For both measures, higher is better.

### 5.4 BASELINES' HYPERPARAMETERS

For ODIN (Liang et al., 2018), we set the temperature to 1000 and the noise magnitude to zero. We take a temperature equal to one for Energy (Liu et al., 2020). We set the temperature to one for GradNorm (GradN. for short; Huang et al., 2021). For Mahalanobis (Maha. for short; Lee et al., 2018), we take only the scores of the outputs of the penultimate layer of the network. The MSP (Hendrycks & Gimpel, 2017) does not have any hyperparameters. For ReAct (Sun et al., 2021), we compute the activation clipping threshold with a percentile equals to 90. For KNN (Sun et al., 2022) we set as hyperparameters $\alpha = 1\%$ and $k = 10$. Our method is hyperparameter free.

## 6 RESULTS AND DISCUSSION

We report our main results in Table 1, which includes the performance for the three model architectures, four OOD datasets, and seven detection methods. Our results are across the board consistent and on average superior to previous methods, obtaining an AUROC of 90% on average (see Table 1). We also ran experiments on a CIFAR-10 benchmark, showing that our method also achieves great performance for the small image datasets. Results are delegated to the Appendix (see Section A.4).

**Multivariate OOD Scores is Not Enough.** Even though well-known multivariate novelty (or anomaly) detection techniques, such as One-class SVM (Cortes & Vapnik, 1995), Isolation Forest (Liu et al., 2008), Extended Isolation Forest (Hariri et al., 2021), Local Outlier Factor (LOF; Breunig et al., 2000), k-NN approaches (Fix & Hodges, 1989) and distance-based approaches (Mahalanobis, 1936) are adapted to various scenarios, they showed to be inefficient for integrating layers' informa-

Table 1: Comparison against post-hoc state-of-the-art methods for OOD detection on the ImageNet benchmark. M+Ours stands for using the Mahalanobis distance-based layer score with our proposed unsupervised score aggregation algorithm based on trajectory similarity. Values are in percentage.

| | | iNaturalist | | SUN | | Places | | Textures | | Average | |
|---|---|---|---|---|---|---|---|---|---|---|---|
| | | TNR | ROC | TNR | ROC | TNR | ROC | TNR | ROC | TNR | ROC |
| **BiT-S-101** | MSP | 35.9 | 87.9 | 28.8 | 81.9 | 21.3 | 79.3 | 21.9 | 77.5 | 27.0 | 81.7 |
| | ODIN | 29.3 | 86.7 | 36.7 | 86.8 | 26.0 | 82.7 | 24.1 | 79.3 | 29.0 | 83.9 |
| | Energy | 25.0 | 84.5 | 39.8 | 87.3 | 27.2 | 82.7 | 24.2 | 78.8 | 29.0 | 83.3 |
| | Mahalanobis | 16.5 | 78.3 | 13.2 | 74.5 | 10.5 | 69.6 | **86.6** | **97.3** | 31.7 | 79.9 |
| | GradNorm | 41.3 | 86.0 | 55.2 | 88.2 | 39.0 | **83.3** | 41.2 | 81.0 | 44.2 | 84.6 |
| | ReAct | 46.2 | 88.9 | 10.7 | 65.9 | 7.0 | 62.0 | 8.5 | 65.8 | 18.1 | 70.7 |
| | KNN | 39.7 | 88.9 | 22.1 | 77.5 | 20.6 | 75.9 | 54.1 | 89.7 | 34.1 | 83.0 |
| | Ours | **67.0** | **91.7** | **59.5** | **89.4** | **40.8** | 82.3 | 85.1 | 96.7 | **63.1** | **90.0** |
| **DenseNet-121** | MSP | 50.7 | 89.1 | 33.0 | 81.5 | 30.8 | 81.1 | 32.9 | 79.2 | 36.9 | 82.7 |
| | ODIN | 60.4 | 92.8 | 45.2 | 87.0 | 40.3 | 85.1 | 45.3 | 85.0 | 47.8 | 87.5 |
| | Energy | 60.3 | 92.7 | 48.0 | 87.4 | 42.2 | 85.2 | 47.9 | 85.4 | 49.6 | 87.7 |
| | Mahalanobis | 3.5 | 59.7 | 4.8 | 57.0 | 4.6 | 54.8 | 54.4 | 88.3 | 16.8 | 64.9 |
| | GradNorm | 73.3 | 93.4 | 59.1 | 88.8 | 48.0 | 84.1 | 56.7 | 87.7 | 59.3 | 88.5 |
| | ReAct | 68.8 | 93.9 | 51.3 | 89.6 | 44.5 | 86.6 | 48.1 | 87.6 | 53.2 | 89.4 |
| | KNN | 57.1 | 92.1 | 33.2 | 83.6 | 26.8 | 79.6 | 81.5 | 96.5 | 49.7 | 88.0 |
| | Ours | 65.7 | 92.8 | **68.0** | **92.1** | 52.4 | 87.3 | 88.3 | 97.5 | 68.6 | 92.4 |
| | ReAct+Ours | **80.4** | **96.4** | 62.2 | 91.8 | 52.3 | **88.0** | 81.8 | 96.5 | **69.2** | **93.2** |
| **ViT-B-16** | MSP | 48.5 | 88.2 | 33.5 | 80.9 | 31.3 | 80.4 | 39.8 | 83.0 | 38.3 | 83.1 |
| | ODIN | 49.9 | 86.0 | 31.5 | 75.2 | 33.7 | 76.5 | 42.6 | 81.2 | 39.4 | 79.7 |
| | Energy | 35.9 | 79.2 | 27.2 | 70.2 | 25.7 | 68.4 | 41.5 | 79.3 | 32.6 | 74.3 |
| | Mahalanobis | **81.2** | **96.0** | 40.7 | **85.3** | **40.0** | **84.2** | 46.3 | 87.5 | 52.1 | 88.2 |
| | GradNorm | 53.5 | 91.2 | **41.5** | 85.3 | 38.3 | 83.4 | 48.1 | 86.5 | 45.3 | 86.6 |
| | ReAct | 33.4 | 85.5 | 26.9 | 78.8 | 25.6 | 77.3 | 42.2 | 84.5 | 32.0 | 81.5 |
| | KNN | 41.0 | 88.9 | 17.8 | 79.4 | 18.2 | 77.7 | 44.0 | 87.8 | 30.3 | 83.4 |
| | Ours | 58.6 | 93.3 | 32.2 | 82.1 | 31.5 | 80.7 | 56.7 | 91.1 | 44.7 | 86.8 |
| | M+Ours | 77.9 | 95.5 | 34.8 | 83.2 | 32.6 | 81.4 | **79.1** | **94.9** | **56.1** | **88.8** |

tion. A hypothesis that explains this failure is the important sequential dependence pattern we noticed in the in-distribution layer-wise scores. Table 2 shows the performance of a few unsupervised aggregation methods based on a multivariate OOD detection paradigm. We tried typical methods: evaluating the Euclidean and Mahalanobis distance w.r.t the training set Gaussian representation, fitting an Isolation Forest, and a One-class SVM on training trajectory vectors. We compare the results with the performance of taking *only* the penultimate layer scores, and we observe that the standard multivariate aggregation fails to improve the scores for DenseNet-121.

Table 2: The first two rows show single-layer baselines where *Penultimate layer* is our layer score on the penultimate layer outputs. The subsequent rows show the performance of other unsupervised multivariate aggregation methods and our method. We ran experiments with a DenseNet-121 model.

| | iNaturalist | | SUN | | Places | | Textures | | Average | |
|---|---|---|---|---|---|---|---|---|---|---|
| | TNR | ROC | TNR | ROC | TNR | ROC | TNR | ROC | TNR | ROC |
| Grad Norm | 73.3 | 93.4 | 59.1 | 88.8 | 48.0 | 84.1 | 56.7 | 87.7 | 59.3 | 88.5 |
| Penultimate layer (Ours) | **78.9** | **95.2** | 61.9 | 90.0 | 51.6 | 86.0 | 68.8 | 90.7 | 65.3 | 90.5 |
| Euclidian distance | 63.7 | 88.8 | 53.6 | 84.8 | 40.8 | 78.4 | 83.5 | 95.6 | 60.4 | 86.9 |
| Mahalanobis distance | 40.1 | 82.8 | 26.6 | 74.5 | 20.8 | 69.4 | 73.5 | 93.5 | 40.3 | 80.1 |
| Isolation Forest | 60.3 | 87.6 | 46.9 | 82.6 | 36.3 | 76.7 | 81.0 | 95.3 | 56.1 | 85.6 |
| One Class SVM | 64.2 | 89.0 | 54.0 | 85.0 | 41.2 | 78.6 | 83.8 | 95.7 | 60.8 | 87.0 |
| Trajectory Proj. (Ours) | 65.7 | 92.8 | **68.0** | **92.1** | **52.4** | **87.3** | **88.3** | **97.5** | **68.6** | **92.4** |

**Qualitative Evaluation of the Functional Dataset.** The test in-distribution trajectories follow a well-defined trend very similar to the training distribution (see Figure 3a). While the OOD trajectories manifest mainly as shape and magnitude anomalies (w.r.t. the taxonomy of Hubert et al., 2015). These characteristics reflects on the histogram of our detection score (see Figure 3b). The in-distribution histogram is generally symmetric, while the histogram for the OOD data is typically skewed and has a smaller average. Please refer to the Appendix A.5 for additional similar figures.

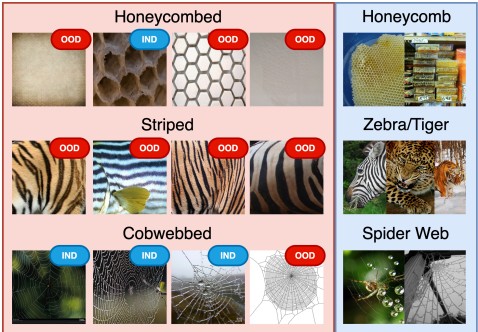
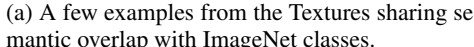
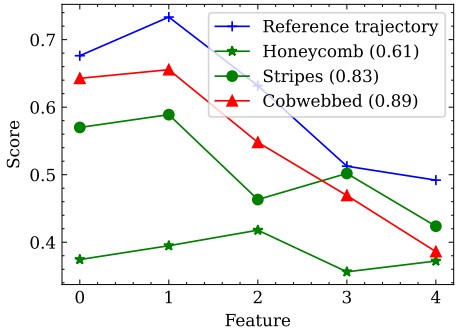

(a) A few examples from the Textures sharing semantic overlap with ImageNet classes.

(b) Trajectories of the leftmost examples of Fig. 4a and their OOD scores in parenthesis.

Figure 4: Study case on OOD detection of individual samples for our method on classes of semantic overlap between the ImageNet and Textures datasets. The badge on each image on Fig. 4a shows the label given by our OOD binary discriminator. We set as threshold the score value with 95% TPR.

**Study Case.** There are a few overlaps in terms of the semantics of class names in the Textures and ImageNet datasets. In particular, "honeycombed" in Textures versus "honeycomb" in ImageNet, "stripes" vs. "zebra", "tiger", and "tiger cat", and "cobwebbed" vs. "spider web". We showed that our method decreases the false negatives in this OOD benchmark. In order to better understand how our method can discriminate where baselines often fail, we designed a simple study case. Take the Honeycombed vs. Honeycomb, for instance (the first row of Fig. 4a shows four examples of this partition and a couple of examples from ImageNet). The honeycomb from ImageNet references natural honeycombs, usually found in nature, while honeycombed have a wider definition attached to artificial patterns. In this class, the Energy baseline makes 108 false negative mistakes, while we only make 20 mistakes. We noticed that some of our mistakes are aligned with real examples of honeycombs (e.g., the second example from the first row), whilst we confidently classify other patterns correctly as OOD. For the Spider webs class, most examples from Textures are visually closer to ImageNet. For the striped case (middle row), our method flags only 16 examples as being in-distribution, but we noticed an average higher score for the trajectories in Fig. 4b. Note that, for the animal classes, the context and head are essential features for classifying them. Overall, the study shows that our scores are aligned with the semantic proximity between testing samples and the training set.

**Potential Shortcomings of our Method.** We believe this work is only the first step in paving the way for efficient post-score aggregation as we have tackled an open and challenging problem of combining multi-layer information. However, we believe there is room for improvement since our metric lives in the inner product space, which is a specific case for more general structures found in Hilbert spaces that might contain more adequate metrics. Another concern that may arise is that, from a practical point of view, current third-party ML services often restrict the practitioner from accessing the intermediate outputs of the model in production. Nonetheless, the service provider has access to this information and could leverage it to deliver OOD detection as a service, for instance.

## 7 CONCLUSION

In this work, we introduced an original approach to OOD detection based on a functional view of the pre-trained multi-layer neural network that leverages the sample's trajectories through the layers. We shifted the problem from a multivariate to a functional perspective in a simple yet original way. Our method detects samples whose trajectories differ from the typical behavior characterized by the training set. The key ingredient relies on the statistical dependencies of the scores extracted at each layer, using a purely self-supervised algorithm. Beyond the novelty and practical advantages of the algorithm, our results establish the value of using a functional approach as an unsupervised technique to combine multiple scores, which offers an exciting alternative to usual single-layer detection methods. We hope this work will be a first step to pave a new way for future research to enhance the safety of AI systems.

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

# A APPENDIX

## A.1 ALGORITHM AND COMPUTATIONAL DETAILS

This section introduces further details on the computation algorithm and resources. Algorithms 1 and 2 describe how to extract the neural functional representations from the samples and compute the OOD score from test samples, respectively. Note that we emphasize the "functional representations", because the global behavior of the trajectory matters. In very basic terms, the method can look into the past and future of the series, contrary to a "sequential point of view" which is restricted to the past of the series only, not globally on the trajectory.

---

**Algorithm 1:** Neural functional representation extraction algorithm computed offline.

**Input** : Training dataset $\mathcal{S}_N = \{(\boldsymbol{x}_i, y_i)\}_{i=1}^N$ and a pre-trained DNN $f = f_{L+1} \circ \cdots \circ f_1$.
**Output:** Reference trajectory $\bar{\boldsymbol{u}}$, scaling vector $\max(\mathcal{U})$, and neural functional trajectory
      extraction function $\phi(\cdot; \{\boldsymbol{\mu}_{\ell,y}: y \in \mathcal{Y} \text{ and } \ell \in \{1, \ldots, L+1\}\})$.

```
// Training dataset feature extraction
```
**for** $\ell \in \{1, \ldots, L+1\}$ **do**
    $\boldsymbol{z}_{\ell,i} \leftarrow (f_\ell \circ \cdots \circ f_1)(\boldsymbol{x}_i)$
    **for** $y \in \mathcal{Y}$ **do**
        $\boldsymbol{\mu}_{\ell,y} \leftarrow \frac{1}{N_y} \sum_{i=1}^{N_y} \boldsymbol{z}_{\ell,i}$   `// Class conditional features prototypes`
    **end**
**end**
**for** $i \in \{1, \ldots, N\}$ **do**
    `// Functional trajectory extractor` $\phi(\cdot)$
    $\boldsymbol{u}_i \leftarrow [\mathrm{d}_1(\boldsymbol{z}_{1,i}; \{\boldsymbol{\mu}_{1,y}: \forall y \in \mathcal{Y}\}), \ldots, \mathrm{d}_{L+1}(\boldsymbol{z}_{L+1,i}; \{\boldsymbol{\mu}_{L+1,y}: \forall y \in \mathcal{Y}\})]$
**end**
$\mathcal{U} \leftarrow \{\boldsymbol{u}\}_{i=1}^N$
$\max(\mathcal{U}) \leftarrow [\max_i u_{i,1}, \ldots, \max_i u_{i,L+1}]^\top$
$\bar{\boldsymbol{u}} = \frac{1}{N} \sum_{i=1}^N \frac{\boldsymbol{u}_i}{\max(\mathcal{U})}$
**return** $\bar{\boldsymbol{u}}$, $\max(\mathcal{U})$, $\phi(\cdot; \{\boldsymbol{\mu}_{\ell,y}: y \in \mathcal{Y} \text{ and } \ell \in \{1, \ldots, L+1\}\})$

---

**Algorithm 2:** Out-of-distribution score computation online.

**Input** : Test sample $\boldsymbol{x}_0$, the DNN $f = f_{L+1} \circ \cdots \circ f_1$, reference trajectory $\bar{\boldsymbol{u}}$, scaling vector
      $\max(\mathcal{U})$, and neural functional trajectory extraction function
      $\phi(\cdot; \{\boldsymbol{\mu}_{\ell,y}: y \in \mathcal{Y} \text{ and } \ell \in \{1, \ldots, L+1\}\})$.
**Output:** Test sample's OOD score $s(\boldsymbol{x}_0)$.

**for** $\ell \in \{1, \ldots, L+1\}$ **do**
    $\boldsymbol{z}_{\ell,0} \leftarrow (f_\ell \circ \cdots \circ f_1)(\boldsymbol{x}_0)$
**end**
$\boldsymbol{u}_0 \leftarrow \phi(\boldsymbol{z}_0; \{\boldsymbol{\mu}_{\ell,y}: y \in \mathcal{Y} \text{ and } \ell \in \{1, \ldots, L+1\}\})$
$\tilde{\boldsymbol{u}}_0 \leftarrow \frac{\boldsymbol{u}_0}{\max(\mathcal{U})}$
$s(\boldsymbol{x}_0) \leftarrow \frac{1}{\|\bar{\boldsymbol{u}}\|^2} \sum_{i=1}^{L+1} \tilde{\boldsymbol{u}}_{0,i} \bar{\boldsymbol{u}}_i$
**return** $s(\boldsymbol{x}_0)$

---

### A.1.1 COMPUTING RESOURCES

We run our experiments on an internal cluster. Since we use pre-trained popular models, it was not necessary to retrain the deep models. Thus, our results should be reproducible with a single GPU. Since we are dealing with ImageNet datasets (approximately 150GB), a large storage capacity is expected. We save the features in memory to speed up downstream analysis of the algorithm, which may occupy around 200GB of storage.

In this section, we conduct a time analysis of our algorithm. It is worth noting that most of the calculation burden is done offline. At inference, only a forward pass and feature-wise scores are computed. We conducted a practical experiment where we performed live inference and OOD computation with three models for the MSP, Energy, Mahalanobis, and Trajectory Projection (ours) methods. The results normalized by the inference time are available in Table 3 below. We reckon that there may exist better computationally efficient implementations of these algorithms. So this remains a naive benchmark of their computational overhead.

Table 3: Batch runtime for OOD detection methods normalized by the time of one forward pass.

|  | Forward pass | MSP | Energy | Mahalanobis | Ours |
|---|---|---|---|---|---|
| BiT-S-101 | 1.00 | 1.00 | 1.00 | 1.21 | 1.19 |
| DenseNet-121 | 1.00 | 1.00 | 1.01 | 1.54 | 1.61 |
| ViT-B-16 | 1.00 | 1.01 | 1.05 | 2.12 | 2.15 |

## A.2  LATENT FEATURES

### A.2.1  BIT-S-101

We used the outputs of the following features to calculate our scores for the BiT-S-101 model:

```
[body.block1, body.block2, body.block3, body.block4, head.flatten]
```

### A.2.2  DENSENET-121

We used the outputs of the following features to calculate our scores for the DenseNet-121 model:

```
[features.transition1.pool, features.transition2.pool,
features.transition3.pool, features.norm5, flatten, classifier]
```

### A.2.3  VIT-B-16

We used the outputs of the following features to calculate our scores for the ViT-B-16 model:

```
[encoder.layers.encoder_layer_0, encoder.layers.encoder_layer_1,
encoder.layers.encoder_layer_2, encoder.layers.encoder_layer_3,
encoder.layers.encoder_layer_4, encoder.layers.encoder_layer_5,
encoder.layers.encoder_layer_6, encoder.layers.encoder_layer_7,
encoder.layers.encoder_layer_8, encoder.layers.encoder_layer_9,
encoder.layers.encoder_layer_10, encoder.layers.encoder_layer_11,
encoder.ln, getitem_5, heads.head]
```

### A.2.4  INTRA-BLOCK CONVOLUTIONAL LAYERS ABLATION STUDY

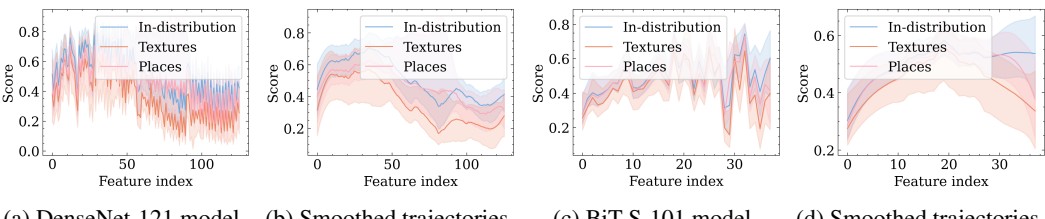

(a) DenseNet-121 model.  (b) Smoothed trajectories.  (c) BiT-S-101 model.  (d) Smoothed trajectories.

Figure 5: Trajectories of scores from every intermediate convolutional layer.

We showed through several benchmarks that taking the outputs of each convolutional block for the DenseNet-121 and BiT-S-101 models is enough to obtain excellent results. We conduct further

ablation studies to understand the impact of adding the intra-blocks convolutional layers in the trajectories. To do so, we extracted the outputs of every intermediate convolutional output of these networks, and we plotted their score trajectories in Figure 5. The resulting trajectories are noisy and would be unfitting to represent the data properly. As a solution, we propose smoothing the curves with a polynomial filter to extract more manageable trajectories. Also, we observe regions of high variance for the in-distribution trajectory inside block 2 for the DenseNet-121 model (see Fig. 5b). A further preprocessing could be simply filtering out these features of high variance that would be unreliable for the downstream task of OOD detection.

## A.3 ON THE CENTRALITY OF THE CLASS-CONDITIONAL FEATURES MAPS

In this section, we study whether the mean vectors are sufficiently informative for statistically modelling the class-conditional embedding features. From a statistical point of view, the average would be informative if the data is compact. To address this point, we plotted the median and the mean for the coordinates of the feature map and measured their difference in Figure 7. We observed that they almost superpose in most dimensions or are separated by a minor difference, which indicates that the data is compact and central. In addition, we showed in Figure 6 that the halfspace data depth (Dyckerhoff & Mozharovskyi, 2014) of the mean vector of a given is superior to the maximum depth of a training sample vector of the same class, suggesting the average is central or deep in the feature data distribution. From a practical point of view, the clear advantage of using only the mean as a reference are computational efficiency, simplicity, and interpretability. We believe that future work directions could be exploring a method that better models the density in the embedding features, especially as more accurate classifiers are developed.

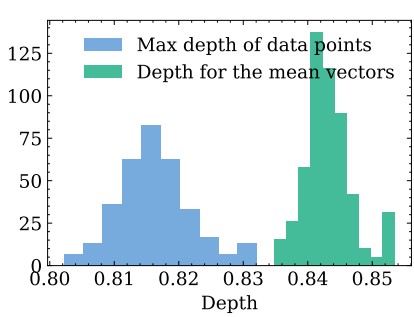

Figure 6: Histogram showing that the halfspace depth of the average vectors for a given class is higher than the highest depth of an embedding feature vector of the same class, demonstrating multivariate centrality.

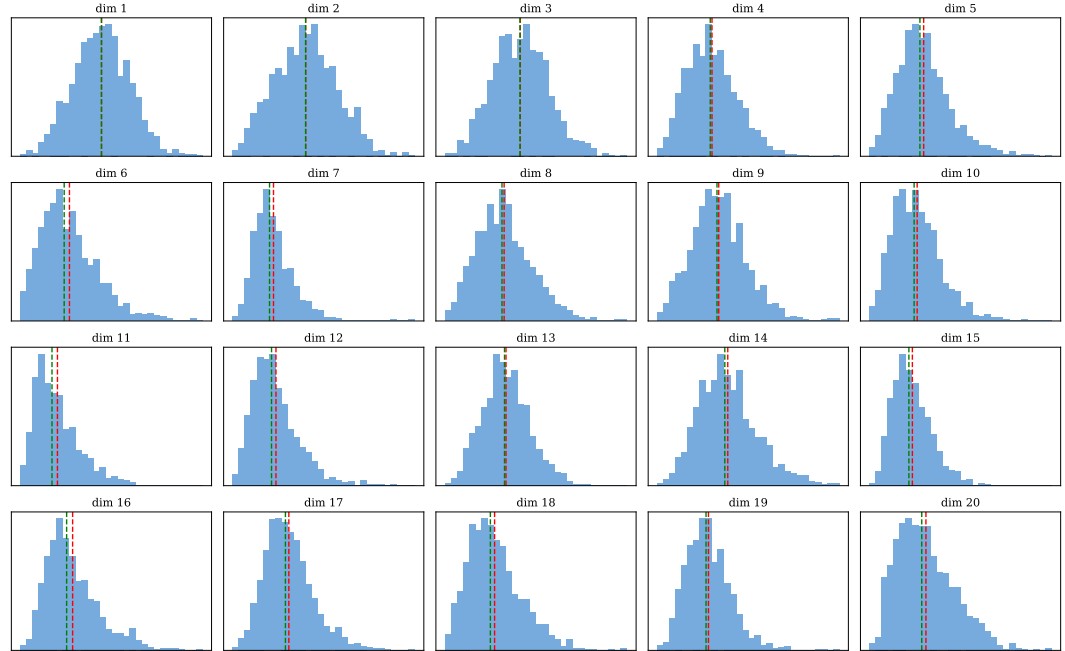

Figure 7: Histogram for the 20 first dimensions of the penultimate feature of a DenseNet-121 for class index 0 of ImageNet. The green line is the average and the red line is the estimated median.

## A.4  RESULTS ON CIFAR-10

We ran experiments with a ResNet-18 model trained on CIFAR-10 and evaluate the OOD performance of a few method compared to ours. We extract features from the outputs of blocks 2 to 4, the penultimate layer and logits. The results are displayed in the table below. Our method outperforms comparable state-of-the-art methods by 2.4% on average AUROC, demonstrating that it is consistent and suitable for OOD detection on small datasets too.

Table 4: CIFAR-10 benchmark results in terms of AUROC based on a ResNet-18 model.

|  | MSP | ODIN | Energy | KNN | ReAct | Ours |
|---|---|---|---|---|---|---|
| CIFAR-100 | 88.0 | 88.8 | 89.1 | 89.8 | 89.7 | 89.4 |
| SVHN | 91.5 | 91.9 | 92.0 | 94.9 | 94.6 | 99.0 |
| LSUN (c) | 95.1 | 98.5 | 98.9 | 97.0 | 97.9 | 99.8 |
| LSUN (r) | 92.2 | 94.9 | 95.3 | 95.8 | 96.7 | 99.8 |
| TinyImageNet | 89.8 | 91.1 | 91.7 | 92.8 | 93.8 | 98.0 |
| Places-365 | 90.1 | 92.9 | 93.2 | 93.7 | 94.7 | 93.6 |
| Textures | 88.5 | 86.4 | 87.2 | 94.2 | 93.4 | 97.9 |
| Average | 90.7 | 92.1 | 92.5 | 94.0 | 94.4 | **96.8** |

## A.5 ADDITIONAL PLOTS, FUNCTIONAL DATASET, HISTOGRAMS AND ROC CURVES

We display additional plots in Figures 9 and 8 for the observed functional data, the histogram of our scores showing separation between in-distribution and OOD data and the ROC curves for all of our experiments.

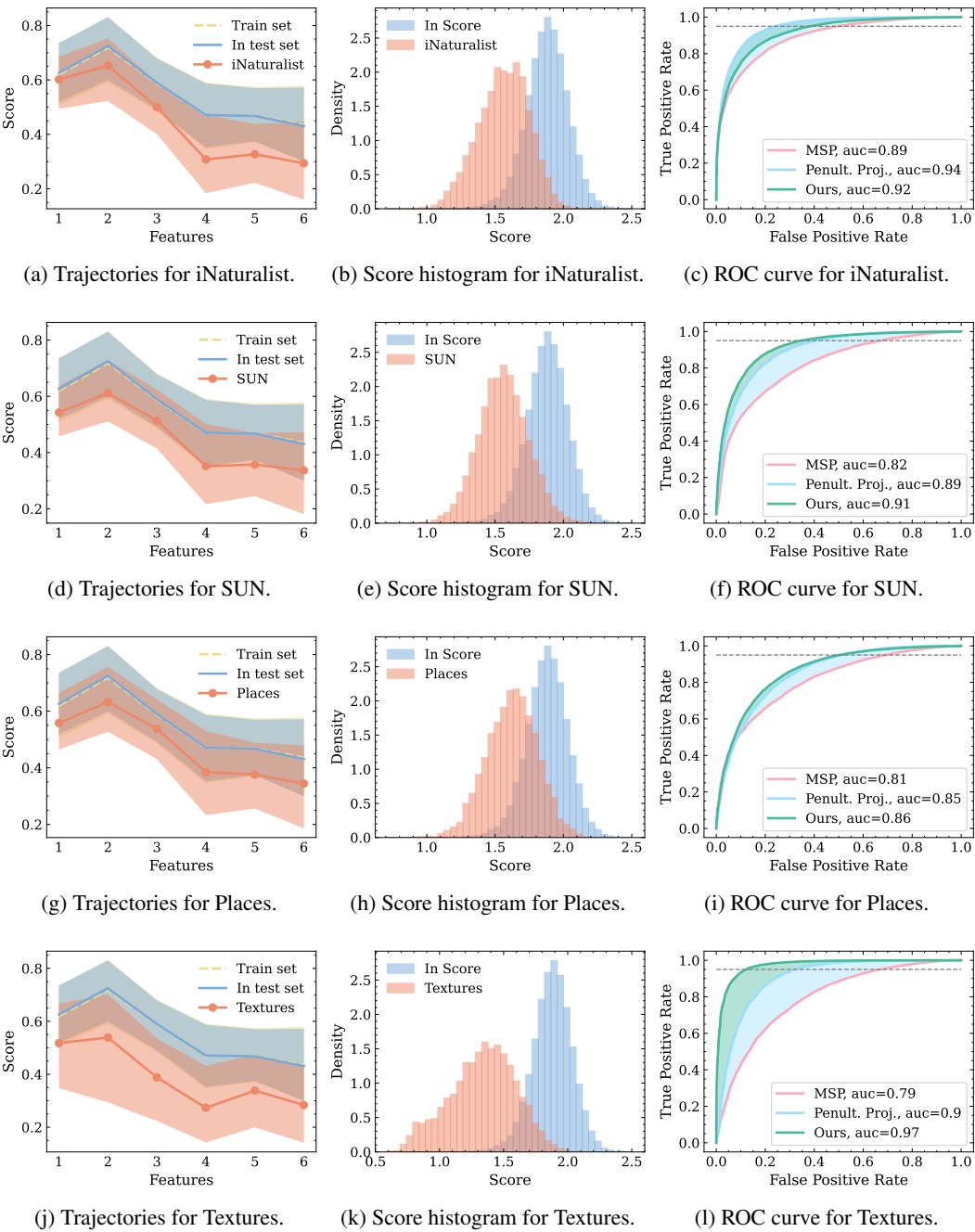

(a) Trajectories for iNaturalist.  (b) Score histogram for iNaturalist.  (c) ROC curve for iNaturalist.

(d) Trajectories for SUN.  (e) Score histogram for SUN.  (f) ROC curve for SUN.

(g) Trajectories for Places.  (h) Score histogram for Places.  (i) ROC curve for Places.

(j) Trajectories for Textures.  (k) Score histogram for Textures.  (l) ROC curve for Textures.

Figure 8: Average trajectories, OOD detection score histogram and ROC curve for the DenseNet-121 model.

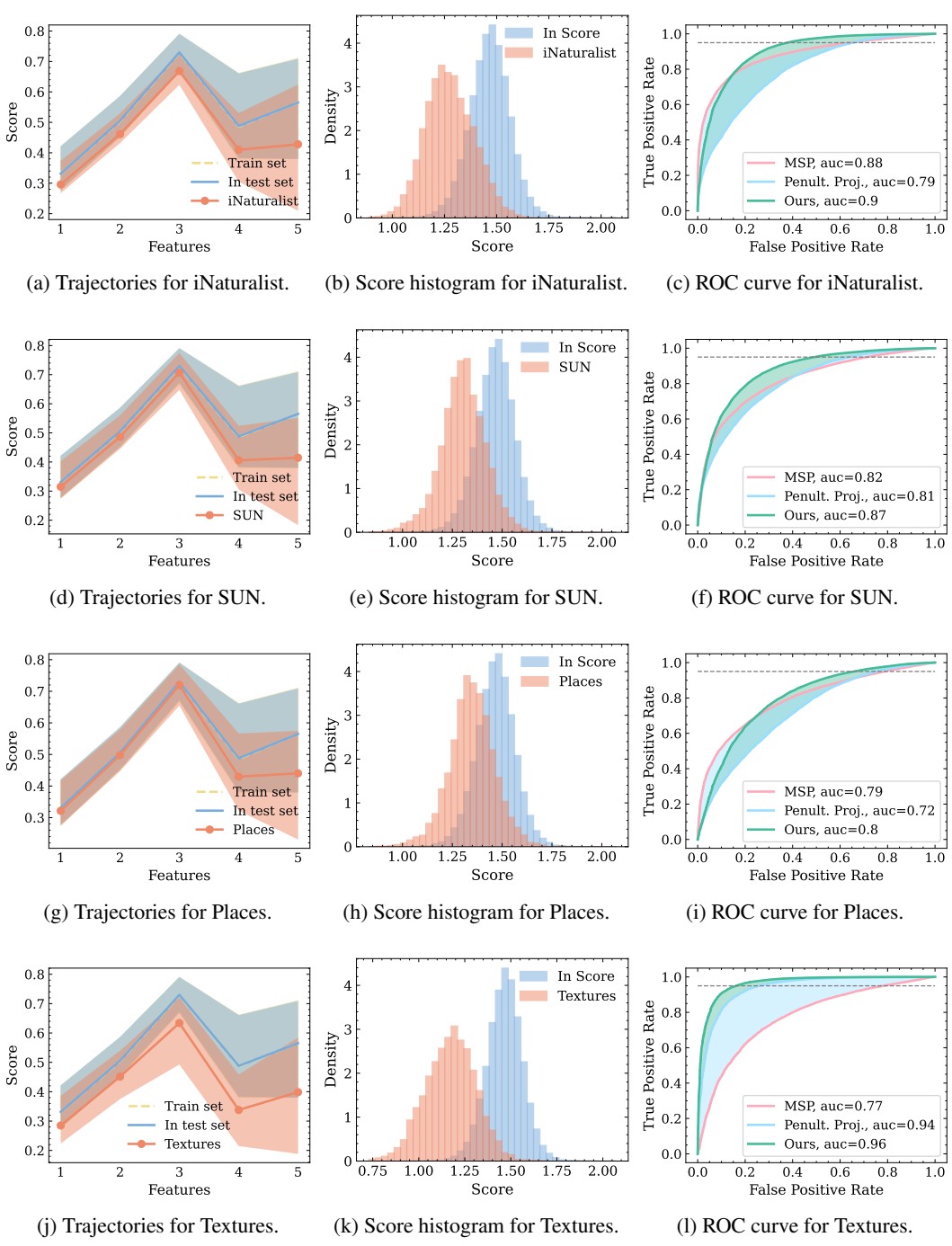

(a) Trajectories for iNaturalist. (b) Score histogram for iNaturalist. (c) ROC curve for iNaturalist.

(d) Trajectories for SUN. (e) Score histogram for SUN. (f) ROC curve for SUN.

(g) Trajectories for Places. (h) Score histogram for Places. (i) ROC curve for Places.

(j) Trajectories for Textures. (k) Score histogram for Textures. (l) ROC curve for Textures.

Figure 9: Average trajectories, OOD detection score histogram and ROC curve for the BiT-S-101 model.

## A.6 DATASETS

Figure 10 shows a few examples from the in-distribution and out-of-distributions datasets. We also list below the subsets of classes used for the OOD datasets following Huang & Li (2021):

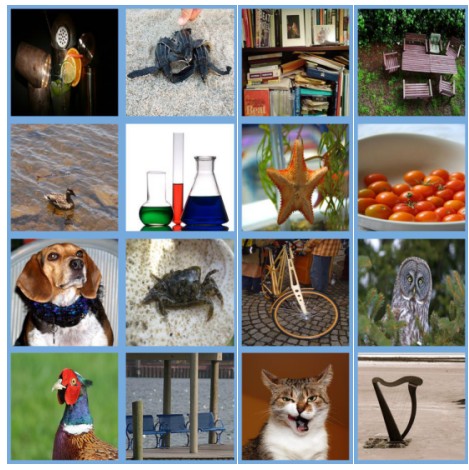
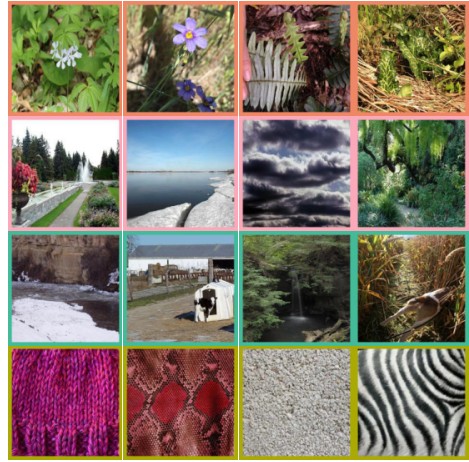

|  |  |
|:---:|:---:|
| (a) In-distribution. | (b) Out-of-distribution. |

Figure 10: Example of data samples from in-distribution (ImageNet) and OOD samples from iNaturalist, SUN, Places365 and Textures datasets.

### A.6.1 iNATURALIST

The classes considered for the iNaturalist dataset were:

```
[Coprosma lucida, Cucurbita foetidissima, Mitella diphylla, Selaginella
bigelovii, Toxicodendron vernix, Rumex obtusifolius, Ceratophyllum
demersum, Streptopus amplexifolius, Portulaca oleracea, Cynodon dactylon,
Agave lechuguilla, Pennantia corymbosa, Sapindus saponaria, Prunus
serotina, Chondracanthus exasperatus, Sambucus racemosa, Polypodium
vulgare, Rhus integrifolia, Woodwardia areolata, Epifagus virginiana,
Rubus idaeus, Croton setiger, Mammillaria dioica, Opuntia littoralis,
Cercis canadensis, Psidium guajava, Asclepias exaltata, Linaria purpurea,
Ferocactus wislizeni, Briza minor, Arbutus menziesii, Corylus americana,
Pleopeltis polypodioides, Myoporum laetum, Persea americana, Avena
fatua, Blechnum discolor, Physocarpus capitatus, Ungnadia speciosa,
Cercocarpus betuloides, Arisaema dracontium, Juniperus californica,
Euphorbia prostrata, Leptopteris hymenophylloides, Arum italicum,
Raphanus sativus, Myrsine australis, Lupinus stiversii, Pinus echinata,
Geum macrophyllum, Ripogonum scandens, Echinocereus triglochidiatus,
Cupressus macrocarpa, Ulmus crassifolia, Phormium tenax, Aptenia
cordifolia, Osmunda claytoniana, Datura wrightii, Solanum rostratum,
Viola adunca, Toxicodendron diversilobum, Viola sororia, Uropappus
lindleyi, Veronica chamaedrys, Adenocaulon bicolor, Clintonia uniflora,
Cirsium scariosum, Arum maculatum, Taraxacum officinale officinale,
Orthilia secunda, Eryngium yuccifolium, Diodia virginiana, Cuscuta
gronovii, Sisyrinchium montanum, Lotus corniculatus, Lamium purpureum,
Ranunculus repens, Hirschfeldia incana, Phlox divaricata laphamii, Lilium
martagon, Clarkia purpurea, Hibiscus moscheutos, Polanisia dodecandra,
Fallugia paradoxa, Oenothera rosea, Proboscidea louisianica, Packera
glabella, Impatiens parviflora, Glaucium flavum, Cirsium andersonii,
Heliopsis helianthoides, Hesperis matronalis, Callirhoe pedata, Crocosmia
crocosmiiflora, Calochortus albus, Nuttallanthus canadensis, Argemone
albiflora, Eriogonum fasciculatum, Pyrrhopappus pauciflorus, Zantedeschia
aethiopica, Melilotus officinalis, Peritoma arborea, Sisyrinchium bellum,
Lobelia siphilitica, Sorghastrum nutans, Typha domingensis, Rubus
laciniatus, Dichelostemma congestum, Chimaphila maculata, Echinocactus
texensis]
```

### A.6.2 SUN

The classes considered for the SUN dataset were:

[badlands, bamboo forest, bayou, botanical garden, canal (natural),
canal (urban), catacomb, cavern (indoor), corn field, creek, crevasse,
desert (sand), desert (vegetation), field (cultivated), field (wild),
fishpond, forest (broadleaf), forest (needleleaf), forest path, forest
road, hayfield, ice floe, ice shelf, iceberg, islet, marsh, ocean,
orchard, pond, rainforest, rice paddy, river, rock arch, sky, snowfield,
swamp, tree farm, trench, vineyard, waterfall (block), waterfall (fan),
waterfall (plunge), wave, wheat field, herb garden, putting green, ski
slope, topiary garden, vegetable garden, formal garden]

### A.6.3 PLACES365

The classes considered for the Places365 dataset were:

[badlands, bamboo forest, canal (natural), canal (urban), corn field,
creek, crevasse, desert (sand), desert (vegetation), desert road, field
(cultivated), field (wild), field road, forest (broadleaf), forest path,
forest road, formal garden, glacier, grotto, hayfield, ice floe, ice
shelf, iceberg, igloo, islet, japanese garden, lagoon, lawn, marsh,
ocean, orchard, pond, rainforest, rice paddy, river, rock arch, ski
slope, sky, snowfield, swamp, swimming hole, topiary garden, tree farm,
trench, tundra, underwater (ocean deep), vegetable garden, waterfall,
wave, wheat field]

### A.6.4 TEXTURES

The classes considered for the Textures dataset were:

[banded, blotchy, braided, bubbly, bumpy, chequered, cobwebbed, cracked,
crosshatched, crystalline, dotted, dibrous, flecked, freckled, frilly,
gauzy, grid, grooved, honeycombed, interlaced, knitted, lacelike,
lined, marbled, matted, meshed, paisley, perforated, pitted, pleated,
polka-dotted, porous, potholed, scaly, smeared, spiralled, sprinkled,
stained, stratified, striped, studded, veined, waffled, woven, wrinkled,
zigzagged]

