# OpenReview forum: "A Functional Perspective on Multi-Layer Out-of-Distribution Detection"
_ICLR.cc/2023/Conference — Submitted to ICLR 2023_

### Official Review · Reviewer_5MWe · 2022-10-23

**Confidence:** 3
**Correctness:** 3
**Technical Novelty And Significance:** 3
**Empirical Novelty And Significance:** 3
**Recommendation:** 6

**Clarity, Quality, Novelty And Reproducibility:**

The paper is clearly written and has its novelty to some extend. Overall, I think the quality of this paper is sufficient, while several issues may require to be solved in the revision. Further, I did not check the reproducibility of the paper.

**Strength And Weaknesses:**

Strength:

- *The novelty of this paper is satisfactory*. The key observation of this paper is that existing OOD detection methods largely overlook the sequential nature of the underlying problem and thus limit the discriminative power of those methods. It motivates the authors to explore the trajectories through the layers and their statistical dependencies in OOD scoring. To me, it is a relatively new and reasonable idea, which may contribute to the community.

Weakness:

- *What does it mean for the term "functional point of view"*. It seems that the authors want to study the sequential behavior of the model outputs, discerning ID and OOD data by their different trajectory through the network. So, to me, I am not sure why the authors exploit the OOD detection problem in the "functional point of view“ instead of the "sequential point of view". Maybe some references or explanations can be helpful.

- *The adopted methodology in modeling the sequential behavior may require further discussion*. To model the sequential behavior of data, I think it might be a direct choice in using the time-series models such as LSTM. I do not fully understand why the adopted method can model the trajectory of model outputs, and why it can be superior over LSTM.

- *Are the mean vectors sufficiently informative for the embedding features?*. It seems that the proposed method distinguishes ID and OOD data by measuring the distances of features to the average features regarding each layer. Then, a nature question is that: is the mean vector sufficient to cover the diversity of ID data in the embedding space. Some justifications or illustration may require here.

- *More experimental results may require*. The authors conduct experiments on ImageNet benchmark, which is a challenging OOD detection setting with large semantic space and complex data features. However, to fully justify the superiority of the proposal, I think the authors should conduct more experiments (e.g., CIFAR benchmarks, Hard OOD detection [1]). Also, more advanced methods should be included in comparison [1,2].

[1] Yiyou Sun, et al. Out-of-distribution Detection with Deep Nearest Neighbors. ICML'22.

[2] Xuefeng Du, et al. VOS: Learning What You Don't Know by Virtual Outlier Synthesis. ICLR'22.

**Summary Of The Paper:**

This paper studies out-of-distribution detection, aiming at making classification models excel at discerning ID and OOD data. It is an important problem for safety-critical applications, and has attracted increasing attention recently. The authors claim that they adopt an original approach based on the functional view of network, which exploits the trajectories through the layers and their statistical dependencies. Specifically, those data whose trajectories differ from ID data are characterized as OOD data. The authors conduct experiments on ImageNet dataset, and the authors claim their superiority over the state-of-the-art methods.

**Summary Of The Review:**

The proposed method discerns ID and OOD data by analyzing the trajectory of model outputs, which is uncovered by much of the previous works. However, there are several issues that may require to be solved, which I think it can improve the clarity and the quality of the paper.

---

> ### Author Response · Authors · 2022-11-16
> **Reply to Reviewer 5MWe**
>
> **We thank the reviewer for acknowledging the strengths of our method and recognizing the novelty in the proposed work that could benefit the scientific community and promote new ways of developing algorithms toward safer AI systems.**
>
> In the following, we provide detailed answers and address each issue reviewer 5MWe raised.
>
> 1. We will clarify the questioning of the reviewer with some intuitions and point out references on the topic. We have also added this remark to the Appendix (please see Section A.1).
>
> > The term "functional point of view" is related to the global behavior of a trajectory,  not necessarily in a sequential manner. In very basic terms, “the model can look into the past and future” of the series. However, from a "sequential point of view", each timestamp can only depend on the past, not globally on the trajectory.
>
> References:
>
> * James O. Ramsay and Bernard W. Silverman, Functional Data Analysis, Springer-Verlag, 2005.
> * Frederic Ferraty and Phillipe Vieu, Nonparametric Functional Data Analysis, Springer-Verlag, 2006.
>
> 2. We thank the reviewer for suggesting using deep time-series models such as the LSTM to tackle the problem of computing trajectory similarities. However, we believe there are a couple of difficulties in adapting the LSTM to this problem. First, the LSTM is conditioned to past points, while our method is not. We could resolve this by using a bi-directional LSTM. The LSTM could be trained to reconstruct training trajectories, which is an interesting suggestion for future work. To promote simplicity and stress the work's main contribution towards the functional point of view of the neural network for OOD detection, we decided to keep the baseline method as simple as possible. We also demonstrated experimentally that our simple baseline already gives good results, and it does not need any training.
>
> 3. We thank the reviewer for the interesting question of whether the mean vectors are sufficiently informative for the embedding features. From a statistical point of view, the average would be informative if the data is compact. To address this point, we plotted the median and the mean for the coordinates of the feature map and measured their difference. We observed that they almost superpose in most dimensions or are separated by a minor difference, which indicates that the data is compact. In addition, we showed that the halfspace data depth [DM2016] of the class conditional mean vectors are superior to the maximum depth of a training sample vector of the same class, suggesting the centrality of the class conditional feature maps. From a practical point of view, the clear advantage of using only the mean as a reference are computational efficiency, simplicity, and interpretability. We believe that future work could explore a method that better models the density in the embedding features, especially as more performant classifiers are developed. We added this analysis to the Appendix (please see Section A.3).
>
> References:
>
> [DM2016] Dyckerhoff, R., & Mozharovskyi, P. (2016). Exact computation of the halfspace depth. Comput. Stat. Data Anal., 98, 19-30.
>
> 4. We thank the reviewer for pointing out interesting related works and suggesting further benchmarks to prove the novelty and efficacy of our work. We ran experiments with [1] and added it to our main benchmark. Concerning reference [2], it needs custom training; hence we believe this work is not directly compared to ours. We added it as related work in Section 2. For both the ImageNet and CIFAR benchmarks, we outperform [1] by a margin on CE-trained models.  We invite the reviewer to check the updated Table 1 of the manuscript and refer to the general comment for details on the implementation of [1] and the results on the CIFAR benchmark, where our method outperforms SOTA methods for the small dataset benchmark too.
>
> References:
>
> [1] Yiyou Sun, et al. Out-of-distribution Detection with Deep Nearest Neighbors. ICML'22.
>
> [2] Xuefeng Du, et al. VOS: Learning What You Don't Know by Virtual Outlier Synthesis. ICLR'22.
>
> **We believe we have answered all of the concerns raised by reviewer 5MWe, and we kindly invite the reviewer to consider increasing their score.**

---

> > ### Comment · Reviewer_5MWe · 2022-11-17
> > **Thanks for the authors' feedback**
> >
> > Many thanks for the authors' feedback. I think I do not have further questions, and I would like to raise my score to 6. Good luck!

---

### Official Review · Reviewer_CjT9 · 2022-10-24

**Confidence:** 4
**Correctness:** 3
**Technical Novelty And Significance:** 3
**Empirical Novelty And Significance:** 2
**Recommendation:** 6

**Clarity, Quality, Novelty And Reproducibility:**

Clarity, Quality, and Reproducibility are good. The paper is well-organized. The experimental evaluation is weak. The key resources and sufficient details are given. I did not run the source code. I think the experimental results are reproducible. Novelty is fair. The proposed detector is derived from a new perspective.

**Strength And Weaknesses:**

Strength:
- This work proposes a novel OOD detection method by using a multi-layer pre-trained classifier.
- The paper is well-written and the technical details are easy to follow.
- The empirical evidence is significant compared with the baseline methods.

Weakness:
- The method is derived from a functional point of view. The authors do not discuss the influence of the depth of the pre-trained classifier. If the pre-trained model only has five layers, is it proper to use five-length vectors to approximate functional data?
- Mood is also a multi-layer OOD detection method and discusses that the image with different data complexity should use feature extractors with different depth. In this work, the authors do not discuss the influence of data complexity and do not compare the proposed method with Mood. Does the summation operation in (8) weaken the sensitivity of the OOD detector?
- Your results in Table 1 outperform the baseline methods. Comparing with recent progress like KNN with ResNet50 pre-training, the improvement is not significant. Furthermore, if you combine your proposed detector with enhancement methods like ReAct, is it possible to further improve the current performance?

Lin, Ziqian, Sreya Dutta Roy, and Yixuan Li. "Mood: Multi-level out-of-distribution detection." Proceedings of the IEEE/CVF Conference on Computer Vision and Pattern Recognition. 2021.


**Summary Of The Paper:**

This work considers the multi-layer structure of a pre-trained classifier and formulates the information forward from a functional view. Then each test input is transformed into a sequence of layer-wise activations. The authors use the projected distance on the in-class typical direction as the measurement of similarity and propose an OOD detection score by summarizing layer-wise distance. Empirical evaluation shows that the proposed method performs well on four ImageNet benchmarks.

**Summary Of The Review:**

I recommend marginal acceptance. My reasons and problems are listed in the Weaknesses section.

---

> ### Author Response · Authors · 2022-11-16
> **Reply to Reviewr CjT9**
>
> **The authors thank the reviewer for the constructive feedback by recognizing the novelty presented, our efforts toward the manuscript's clarity, and our extensive benchmark on ImageNet.**
>
> In the following, we will address the questions of the reviewer.
>
> 1. The reviewer questions whether our method is appropriate for shallower models. From a functional data analysis perspective, sequences of size $\geq$ 2 are appropriated to be treated as functional sequences. We believe that in a scenario where a 5-layer-deep neural network is performant on its task, it could also benefit from our method for the task of OOD detection. To support our claim, we emphasized that we subsampled 5 layers of a deep model (DenseNet-121), and we showed experimentally that short sequences can approximate functional data well to serve for OOD detection. Please refer to table 2 of the manuscript, where we compare unsupervised methods for computing scores from multiple network layers. Our method outperforms the One-Class SVM algorithm, for instance, by 5.4% AUROC on average.
>
> 2. We will answer bullet point number 2 raised by the reviewer in two parts:
>
> * The work Mood referenced by the reviewer makes an interesting connection between the input data complexity and which layer index will perform the best in detection. This is another factor of explainability of OOD samples that could be combined with the trajectories point of view in future work. Said that, Mood requires every layer output to have a trained classifier attached to it to calculate the score once the output index is selected. Because of this particular architectural design and the necessity of a special training procedure, we believe this work is not directly compared to ours, as we do not have any special requirements at the model training level. We added it as a related work in section 2 of the updated manuscript.
> * Regarding the previous equation (8) (now equation (5)), The inner product is a standard metric for calculating sequence similarity. The rescaling by the max training trajectory coordinates promotes a comparable range between trajectory points, which promotes sensitivity to changes in every dimension of the trajectory. We believe this metric is a great baseline as it gives already good results, but that other measures could be envisioned in future works.
>
> 3. We thank the reviewer for pointing out interesting recent work and clues to ameliorate our method by combining it with activation function clipping as suggested in previous work. We ran experiments with both suggestions. We added a comparison to the KNN and ReAct method in our experimental setting. We also ran experiments combining ReAct and our method, showing improvements on a DenseNet model. To sum up, for all three models, our method outperformed both ReAct and KNN methods by a margin. Combining activation clipping of the intermediate blocks of the DenseNet model improved our method by 0.8% on average AUROC. We invite the reviewer to refer to the general comment and updated Table 1 of the manuscript for more details.
>
> **We believe we have addressed the points raised by reviewer CjT9 in the weaknesses section. Accordingly, we kindly ask the reviewer to iterate on it if necessary and eventually raise their score.**

---

> > ### Comment · Reviewer_CjT9 · 2022-11-28
> > **Response**
> >
> > Thank you for your detailed response. The authors answered my questions about the functional perspective and explained the comparison between the proposed method and MOOD. They also added an empirical comparison with KNN and ReAct. Though the results outperform the baseline methods in Table 1, the improvement is still not significant compared to KNN+ or KNN+ with ReAct. Therefore, I will keep my score.

---

> > > ### Author Response · Authors · 2022-12-06
> > > **Response to Reviewer CjT9**
> > >
> > > We thank the reviewer for their comments. Reviewer CjT9 points out that our method is not significantly better than [2]. However, we argue that 1) we are in a different setting and 2) in a comparable scenario we outperform KNN+ and KNN+ with ReAct.
> > >
> > > ### 1) KNN+ cannot be directly compared to our detector as we are in a different setting
> > >
> > > The KNN+ method is based on a model trained with a contrastive loss and requires supervised hyperparameter validation to tune alpha and k (see Figure 2 in [1]), as well as a particular validation procedure to find the activation clipping threshold. This setup is not comparable to our method for two main reasons:
> > >
> > > - our method is hyperparameter free
> > >
> > > - our detector can be plugged into any kind of encoder without retraining.
> > >
> > > ### 2) We outperform both KNN+ and  KNN+ with React when placed in the same scenario
> > >
> > > Despite these considerations, we ran experiments with our method on a ResNet-50 backbone to directly compare to the results from [1] and [2]. We have also included the results of KNN+ReAct [2], DICE+ReAct [1], and Projection+ReAct (ours - tuned on the training set only) in the table below. We find that our method outperforms all the other methods on average.
> > >
> > > |   Method           | iNaturalist |    | SUN |  | Places |  | Textures |  | Average |  |
> > > | ------------ | ----------- | ----- | ------ | -------- | ------- | ----- | ----- | ----- | ----- | ----- |
> > > |              | FPR95       | AUROC | FPR95  | AUROC    | FPR95   | AUROC | FPR95 | AUROC | FPR95 | AUROC |
> > > | DICE   [1]      | **25.63**       | 94.49 | **35.15** | 90.83    | **46.49**  | **87.48** | 31.72 | 90.30 | 34.75 | 90.77 |
> > > | KNN+ [2]        | 30.18       | **94.89** | 48.99  | 88.63    | 59.15   | 84.71 | **15.55** | 95.40 | 38.47 | 90.91 |
> > > | Ours         | 40.06       | 92.10 | 35.32  | **91.73**    | 50.43   | 87.31 | 16.21 | **96.26** | **35.48** | **91.86** |
> > > |  |  |  |  | |  |  |  |  |  |  |
> > > | DICE + ReAct [1] | **18.64**       | **96.24**| 25.45  | 93.94    | **36.86**   | 90.67 | 28.07 | 92.74 | 27.25 | 93.40 |
> > > | KNN + ReAct  [2] | \-          | \-    | \-     | \-       | \-      | \- | \- | \- | 26.45 | 93.76 |
> > > | Ours + ReAct | 24.31       | 95.82 | **23.96**  | **94.54**    | 36.98   | **91.20** | **16.77** | **96.27** | **25.51** | **94.46** |
> > >
> > > In conclusion, our fully unsupervised method achieves better results and beats competing methods by grasping information from multiple latent representations across different layers. Thus, we consider it to be an important contribution to the OOD detection community. As acknowledged by the reviewer, our results are better than the baselines presented in Table 1.
> > >
> > > In light of these considerations, we hope that the reviewer CjT9 will reconsider their evaluation and increase their score.
> > >
> > > ### References:
> > >
> > > [1] Sun, Y. & Li, Y. Dice: Leveraging sparsification for out-of-distribution detection. In ECCV 2022.
> > >
> > > [2] Sun, Y., Ming, Y., Zhu, X., & Li, Y. Out-of-distribution Detection with Deep Nearest Neighbors. ICML 2022.

---

### Official Review · Reviewer_8gLg · 2022-10-25

**Confidence:** 4
**Correctness:** 3
**Technical Novelty And Significance:** 3
**Empirical Novelty And Significance:** 3
**Recommendation:** 5

**Clarity, Quality, Novelty And Reproducibility:**

This paper does not have good clarity. Thus, the quality and novelty are hard to evaluate.

**Strength And Weaknesses:**

Pros:

The author examines the OOD samples during the forward pass through a novel and interesting perspective that takes feature trajectory into account. It might be a more discriminative way to see how the network handles ID and OOD inputs.

Cons:

1. In this paper, the proposed method uses the class prototype \mu_{l,\hat{y}} corresponding to the network predicted label as the reference. But what if the labels predicted by the network are wrong (we know the top-1 accuracy on ILSVRC 2012 is around 80%)? For a misclassified ID sample, should it be considered as OOD data on the wrong class? The authors use hard labels (i.e., argmax(f(x_0))) to obtain reference trajectories for the corresponding category. What about using soft labels to perform weighted sum (\sum_i {score}_{i}\cdot d(z_{l,0}, \mu_{l,i}))? The authors may explore this in their experiments.

2. The authors need to conduct more detailed ablation study. For example, the authors only take the outputs of five stages (body.block1, body.block2, body.block3, body.block4, head.flatten) as latent features. What if the output after each convolutional layer is tracked? Furthermore, I notice that u_0 in Algorithm 2 is already a trajectory similarity between a test sample and the training features. Whether it is necessary to recalculate the similarity between u_{0} and u needs to be explored in the ablation experiment. And it is best for the author to further analyse the reasons for such a design through theory.

3. The paper is unclear that what shape features are used to calculate cosine similarity. If they are (B, C_l, H_l, W_l), does that introduce a lot of computation (numclass\times(L+1)\times B\times C_l\times H_l\times W_l)? I think the authors need to report in detail the additional computational burden introduced by the proposed method, both FLOPs and time-consuming. Kindly remind the author that when implementing the code, you can first select the feature prototype of a specific category by torch.gather, and then calculate the cosine similarity through matrix multiplication or einsum to avoid repeated calculation between input features and all categories.

4. In Table 1, the authors do not compare with the SOTA method, and the performance of the method proposed in this paper has not yet reached the level of SOTA (e.g., KNN[1], DICE[2]). I encourage the authors to further optimize the proposed method during rebuttal for more competitive results. I will consider raising the rating of this paper after the results are further improved.

5. The authors say "State-of-the-art methods treat step (ii) as a supervised learning problem" in Section 3.2, paragraph 1 which needs some citations to determine which methods do so.

Reference

[1] Sun, Y., Ming, Y., Zhu, X., & Li, Y. Out-of-distribution Detection with Deep Nearest Neighbors. In ICML 2022.

[2] Sun, Y., & Li, Y. Dice: Leveraging sparsification for out-of-distribution detection. In ECCV 2022.


**Summary Of The Paper:**

The paper proposed a feature-based method to get OOD scores that measures the trajectory similarities between the test data and the training data. The method first gets the per-class feature prototype \mu_{l,y} at each layer based on training data, and then figures out the reference trajectory by calculating the cosine similarity between the training samples and the prototypes. During the inference time, the authors evaluate the trajectory consistencies of test samples. The experimental results show that the proposed method outperforms the baseline and some recently proposed methods.

**Summary Of The Review:**

The paper proposed a feature-based method to get OOD scores that measures the trajectory similarities between the test data and the training data. The method first gets the per-class feature prototype \mu_{l,y} at each layer based on training data, and then figures out the reference trajectory by calculating the cosine similarity between the training samples and the prototypes. During the inference time, the authors evaluate the trajectory consistencies of test samples. The experimental results show that the proposed method outperforms the baseline and some recently proposed methods. However, this paper does not clearly demonstrate the motivation, making the justification difficult.

---

> ### Author Response · Authors · 2022-11-16
> **Reply to Reviewer 8gLg**
>
> **We acknowledge the reviewer for carefully reading our manuscript, recognizing the novelty of the work, and suggesting valuable changes to our method that improved our detection performance.**
>
> Next, we assess each issue raised by the reviewer:
>
> 1. The reviewer suggests a modification in the feature map projection score. We addressed this issue by implementing the suggestion proposed by the reviewer. **We provide below a table with the results obtained compared to the previous results, showing on average 1% AUROC improvement.** We made appropriate changes to the manuscript to consider the updated method.
>
> |              |          | iNaturalist | SUN | Places | Textures | Average |
> | ------------ | -------- | ----- | ----- | ----- | ----- | ----- |
> | BiT-S-101    | Previous | 89,8  | 87,0  | 79,6  | 96,3  | 88,2  |
> |              | Updated  | **91,7**  | **89,4**  | **82,3**  | **96,7**  | **90,0**  |
> | DenseNet-121 | Previous | 91,6  | 91,3  | 86,2  | 97,3  | 91,6  |
> |              | Updated  | **92,8**  | **92,1**  | **87,3**  | **97,5**  | **92,4**  |
> | ViT-16-B     | Previous | 92,8  | 80,2  | 78,9  | 90,6  | 85,6  |
> |              | Updated  | **93,3** | **82,1**  | **80,7**  | **91,1**  | **86,8**  |
>
> * Addressing the question of whether misclassified ID samples should be considered as OOD data, the authors believe that misclassifications are not out-of-distribution by definition. Misclassified samples express the limited generalization inherent to the classification model because we suppose the test set was sampled from the true distribution $P_{Y|X}$, just like the training set. We also believe that taking the class probability in computing the score mitigates the potential risk of relying on the misclassified label.
>
> 2. We will respond to the two parts of topic 2 raised by the reviewer.
> * The features were selected in a straightforward way which is by taking the output of every building block of the neural networks. Adding intermediate convolutional layers would add more noise to the trajectory and increase the method's computational complexity. However, we believe the ablation study suggested by the reviewer is valid, so we added it to the appendix of the updated manuscript (please see Section A.2.4).
>
> * In Algorithm 2, $u_0$ is the trajectory extracted by computing the layer-level scores for the input test sample $x_0$. Extracting feature scores through multiple points of the network is essential because samples from different distributions might behave differently in the forward pass, and there is little indication of knowing this in advance. The second score is to test if this test trajectory conforms with the training distribution. We choose a simple score that computes the inner product between the rescaled $u_0$ and the rescaled average trajectory $\bar{u}$. The second method of table 2 is our feature-wise score considering only the penultimate feature of a DenseNet-121 model. We demonstrated a gain of around 2% in average AUROC by considering the entire trajectory. On far OOD data, such as the textures dataset, this difference can go up to 7% AUROC. Thus, we believe this work will be a first step to pave a new way for future research to model neural network behavior and enhance the safety of AI systems.
>
> 3. To prevent the dimensions from increasing to computationally infeasible values, we compute a max pooling operation on the convolutional features before applying the cosine similarity, obtaining a tensor of size (B, C_l) for each layer. The dimensions for each resulting feature map are written in section 5.2. Regarding the computational complexity of our method, we thank the reviewer for pointing out possible implementation ameliorations to our code. We emphasize that most of the computation burden is done offline. We conducted a quick time analysis experiment with the three studied methods at inference time. The results normalized by the time of one forward pass are available in the table below and in the Appendix of the updated manuscript (please see Section A.1.2, Table 3).
>
> |              | Forward pass | MSP  | Energy | Mahalanobis | Ours |
> | ------------ | ------------ | ---- | ------ | ----------- | ---- |
> | BiT-S-101    | 1,00         | 1,00 | 1,00   | 1,21        | 1,22 |
> | DenseNet-121 | 1,00         | 1,00 | 1,01   | 1,54        | 1,61 |
> | ViT-B-16     | 1,00         | 1,01 | 1,05   | 2,12        | 2,15 |
>
> We reckon that there may exist more computationally efficient implementations of these algorithms. So this remains a naive benchmark of their computational overhead.
>
> 4. We thank the reviewer for pointing it out. We updated the manuscript with citations of previous which treat the problem of combining scores from multiple layers in a supervised manner (please see Section 3.2 of the updated manuscript).
>
> **We hope we have addressed the reviewer's most concerning points satisfactorily, and we kindly invite reviewer 8gLg to increase their score.**

---

> > ### Comment · Reviewer_8gLg · 2022-12-04
> > **Thanks for your reply**
> >
> > Thanks for the reply. However, I still have some concerns regarding this paper.
> >
> > Re A3: I think this method does not generalize well across different models, the time cost is only 20% more on ResNet but more than double on ViT, which might not be unacceptable for OOD detection methods.
> >
> > Re A4: I agree with reviewer CjT9. This performance does not seem significant enough compared to KNN+. Also, the results reported by Dice [1] seem to be higher than those reported in the paper.
> >
> > I think the empirical results are not good enough. Thus, it is hard to justify the contributions of this paper to OOD detection field.
> >
> > Could the authors show the potential that the proposed methods can outperform the SOTA methods?
> >
> > Currently, SOTA OOD detection methods are not complex, so there should not be much room to use some tricks to improve detection performance. If we conceptually compare two methods (namely, no tricks are used) and method A outperforms method B, it is difficult to justify that the idea behind B is significant to this field.
> >
> > [1] Sun, Y., & Li, Y. Dice: Leveraging sparsification for out-of-distribution detection. In ECCV 2022.

---

> > > ### Author Response · Authors · 2022-12-06
> > > **Response to Reviewer 8gLg**
> > >
> > > Let us thank reviewer 8gLg for their feedback. They underline two remaining weaknesses that prevent them from raising their grade: 1) the speed of our detector and 2) it's lack of significance with respect to Dice, Dice with ReAct [1], KNN+ and KNN+ with ReAct [2].
> > >
> > > In the following, we provide detailed answers to their concerns, which we hope will help them reconsider their opinion about our work.
> > >
> > > ### 1) We have increased the speed of our method, which is now faster or comparable with KNN+ and KNN+ with ReAct
> > >
> > > Thanks to reviewer 8gLg’s remark, we identified a bottleneck in our naive implementation. We have now replaced a for loop by a matrix multiplication, which leads to way better speed performances, as reported in the Table below, where the scores are normalized by the MSP inference time.
> > >
> > > | Backbone       | MSP  | Energy | KNN  | Ours |
> > > | -------------- | ---- | ------ | ---- | ---- |
> > > | BiT-S-101      | 1,00 | 0,74   | 1,17 | 1,02 |
> > > | DenseNet-121   | 1,00 | 1,02   | 1,40 | 1,24 |
> > > | ViT-B-16       | 1,00 | 1,03   | 1,09 | 1,22 |
> > > | ResNet-50      | 1,00 | 1,04   | 1,38 | 1,33 |
> > >
> > > This table clearly demonstrates that the inference time of our method is both fast and consistent across backbones. Although it exhibits a small overhead compared to the inference time of MSP, we believe that this is an acceptable trade-off in terms of the performance improvement it induces.
> > >
> > > We believe this should address reviewer 8gLg concern and would be happy to release this new implementation upon acceptance.
> > >
> > > ### 2) When considering the same scenario, we consistently outperform Dice, KNN+, and KNN+ with ReAct
> > >
> > > We have conducted several experiments to include a comparison with the state-of-the-art methods mentioned by reviewer 8gLg, computed on a ResNet-50 backbone. Even on this new backbone, our method still consistently outperforms the others, as can be observed in the following Table.
> > >
> > > |   Method           | iNaturalist |    | SUN |  | Places |  | Textures |  | Average |  |
> > > | ------------ | ----------- | ----- | ------ | -------- | ------- | ----- | ----- | ----- | ----- | ----- |
> > > |              | FPR95       | AUROC | FPR95  | AUROC    | FPR95   | AUROC | FPR95 | AUROC | FPR95 | AUROC |
> > > | DICE   [1]      | **25.63**       | 94.49 | **35.15** | 90.83    | **46.49**  | **87.48** | 31.72 | 90.30 | 34.75 | 90.77 |
> > > | KNN+ [2]        | 30.18       | **94.89** | 48.99  | 88.63    | 59.15   | 84.71 | **15.55** | 95.40 | 38.47 | 90.91 |
> > > | Ours         | 40.06       | 92.10 | 35.32  | **91.73**    | 50.43   | 87.31 | 16.21 | **96.26** | **35.48** | **91.86** |
> > > |  |  |  |  | |  |  |  |  |  |  |
> > > | DICE + ReAct [1] | **18.64**       | **96.24**| 25.45  | 93.94    | **36.86**   | 90.67 | 28.07 | 92.74 | 27.25 | 93.40 |
> > > | KNN + ReAct  [2] | \-          | \-    | \-     | \-       | \-      | \- | \- | \- | 26.45 | 93.76 |
> > > | Ours + ReAct | 24.31       | 95.82 | **23.96**  | **94.54**    | 36.98   | **91.20** | **16.77** | **96.27** | **25.51** | **94.46** |
> > >
> > > We hope reviewer 8gLg will acknowledge that, together with this additional experiment, our extensive experimental setting clearly demonstrates that we improve over state-of-the-art methods.
> > >
> > > ---
> > >
> > > Let us make a final remark about reviewer 8gLg comment:
> > >
> > > > “Currently, SOTA OOD detection methods are not complex, so there should not be much room to use some tricks to improve detection performance. If we conceptually compare two methods (namely, no tricks are used) and method A outperforms method B, it is difficult to justify that the idea behind B is significant to this field.”
> > >
> > > We agree with the overall suggestion that adding noise to the community is not relevant. That is precisely why we **motivated** our method by showing that 1) there exists relevant signal regarding OOD detection when looking at the hidden layers of networks and 2) that this information should be considered through the lens of **information processing** across networks, that is, with a temporal view. We implemented these ideas and demonstrated that they lead to a consistent improvement over current state-of-the-art. Far from being an “A and B” comparison, we believe we have brought a new idea to the table that should be considered for future research direction. It may not be revolutionary, but we think it deserves the community's attention as it could lead to safer AI systems.
> > >
> > > In conclusion, we addressed the reviewer's main concern by showing that our method can outperform current SOTA methods by simply changing the backbone to a ResNet-50 trained on ImageNet-1k. Accordingly, we thank the reviewer for the opportunity to showcase the robustness of our method, and we kindly invite the reviewer to raise their score.
> > >
> > > ### References:
> > >
> > > [1] Sun, Y. & Li, Y. Dice: Leveraging sparsification for out-of-distribution detection. In ECCV 2022.
> > >
> > > [2] Sun, Y., Ming, Y., Zhu, X., & Li, Y. Out-of-distribution Detection with Deep Nearest Neighbors. ICML 2022.

---

### Official Review · Reviewer_aTaH · 2022-10-26

**Confidence:** 5
**Correctness:** 3
**Technical Novelty And Significance:** 2
**Empirical Novelty And Significance:** 3
**Recommendation:** 5

**Clarity, Quality, Novelty And Reproducibility:**

Clarity can be improved, the proposed approach is somewhat novel. Code is provided.

**Strength And Weaknesses:**

Strengths:

1. Proposed functions perspective for OOD detection is interesting.

2. During the test, the similarity is measured with respect to cluster means ignoring the covariance matrices. This significantly improves the complexity, especially for the Imagenet with 1000 classes.

3. A series of ablation studies are conducted to analyze various aspects of the approach.

4. Authors provide the code to facilitate reproducibility.

Weaknesses:

1. Is considering the centroids alone, without the covariance, reliable to detect OOD samples? Have the authors experimented with a variant where the similarity is computed considering the covariance matrix as in the Mahalabois approach?

2. Authors claim that the state-of-the-art approaches use supervision to train the OOD classifier. This statement is not completely correct as many recent approaches do not consider supervision (OOD samples or pseudo-OODs) during training.

3. In Fig 2 b, it appears that the trajectories across ID and OOD samples are similar at various feature layers. How does this affect the robustness of the approach?

4. Some of the recent state-of-the-art approaches are not compared against, such as

i. Yiyou Sun, Yifei Ming, Xiaojin Zhu, and Yixuan Li. Out-of-distribution detection with deep nearest neighbors. ICML, 2022.
ii. Vikash Sehwag, Mung Chiang, and Prateek Mittal. SSD: A unified framework for self-supervised outlier detection. ICLR, 2021.

5. Authors are encouraged to consider a commonly used experimental setup where CIFAR is used in distribution and SVHN, TinyImageNet, and LSUNs are used as OOD. This will help to compare against other state-of-the-art approaches and further justify the efficacy of the approach.

6. The clarity and writeup can be improved

i. Fig 1 is hard to follow given the description at that stage.
ii. In Fig 2 c, the conclusion is not clear. Only the self-correlation is evident from the plot the correlations between the features from other layers are somewhat non-uniform.

**Summary Of The Paper:**

This paper presents a functional perspective for out-of-distribution (OOD) detection.  The layer-wise features from a deep neural network are considered a trajectory. The class-wise centroids of the trajectories are computed from the training samples. Given a test trajectory, the similarity with respect to the centroids is estimated as a measure for OOD-ness, i.e., lower similarity indicates an OOD sample. Experiments are conducted by considering Imagenet to be in-distribution and various other datasets, such as iNaturalist, SUN, Places, and Textures, as OOD sets.

**Summary Of The Review:**

While the proposed idea of considering feature trajectories is interesting, it requires additional experimental validations to justify the efficacy of the approach. Please address the comments in the weaknesses section.

---

> ### Author Response · Authors · 2022-11-16
> **Reply to Reviewer aTaH**
>
> **Let us thank reviewer aTaH for their careful reading of the manuscript and constructive feedback. We are glad they acknowledge that our functional perspective on the problem of OOD detection is interesting, recognize that our method is computationally efficient, and appreciate our efforts to explain the gains through ablation studies.**
>
> In the following, we provide detailed answers and address the questions asked by the reviewer:
>
> 1. Our method is composed of two steps:
>
> * At the feature map level, we compute a projection of the test sample into the class conditional prototypes. It has some advantages because estimating the covariance matrix at the feature map level is unreliable for a wide class of deep models and is computationally expensive (O(N*d^2)). However, for transformers, we observed that leveraging the Mahalanobis distance scores to build the trajectories gives slightly superior performance (see Table 1).
>
> * At the level of the trajectories, we compute the inner product between the test trajectory and the average training trajectories. We also ran experiments by computing the similarity using the Mahalanobis distance between a test sample and the typical training average vector and empirical covariance matrix. However, this did not help, and it is slower to compute. We updated table 2 to take this experiment into account.
>
> Overall, our contribution is a simple baseline for unsupervisedly combining multiple layers scores by observing the structure of a sample's trajectory through the network. The experimental results motivate the fact that the proposed statistical model is performant for OOD detection by consistently outperforming state-of-the-art methods.
>
> 2. We have updated the manuscript to clarify the statement. Please see the updated section 3.2.
>
> 3. The behavior highlighted by the reviewer is observed in the first layers of the transformer model and for the SUN and Places datasets on the other models. We believe this could be a limitation of the representation learned by the deep classifiers. Regarding the robustness of our method, the inner product given by the previous equation (8) (now equation (5)) would accumulate constant values to OOD and IND samples equally and does not affect the discriminative power, but only the final score range. It is worth noting that other datasets could behave very differently at the beginning of the feature extraction process, even though this is not represented in this benchmark. Our method would be able to capture this. We also believe the trajectories are a step towards explicability in the OOD generalization/detection field because they permit analysis like this.
>
> 4. We thank the reviewer for pointing out these related works. The work i. is a recent post-hoc approach that computes the k-th smallest Euclidean distance w.r.t a testing sample and the training dataset on the level of the penultimate feature map outputs. We add a comparison to this work. Please refer to the general comment for more details. The work ii. Is built on top of a self-supervised learning procedure which involves training a model to create better representations with a contrastive loss. We believe this is out of the scope of this work, and we will not compare against it.
>
> 5. We ran experiments on a CIFAR benchmark with the suggested datasets to address this question. We refer the reviewer to the general comment for further details and results on the CIFAR benchmark, in which we outperform comparable state-of-the-art methods by a 2.4% AUROC margin on average.
>
> 6. We thank the reviewer for their feedback on the writing of the manuscript. We will make improvements to the clarity of the text and overall readability.
>
>
> **We hope our responses address the reviewer’s questions, and we kindly invite reviewer aTaH to iterate on our response and eventually review their score.**

---

### Author Response · Authors · 2022-11-14
**Summary of the main changes**

We would like to thank all reviewers for their valuable comments and efforts in reviewing our paper. We are carefully revising the manuscript by taking into account the reviewer’s comments. The main modifications on the original manuscript will be done in red during the rebuttal period.

In this global review, we will first address the most common points elaborated by the reviewers, and subsequently, we will respond to each suggestion/question individually.

**Additional baselines**

We position ourselves on a post-hoc only OOD detection without _any_ hyperparameter tuning required. We thank the reviewers for pointing out further post-hoc methods to be added to the benchmark.

ReAct [1]: we fit the activation clipping threshold based on a percentile of 90% of randomly sampled 5% of the training set and used the energy OOD detection score on the logits as suggested in [1]. In [1], 2000 samples of the validation set of ImageNet are used to find the threshold. Since our framework doesn’t need access to any unlabelled validation set to set any hyperparameter, we chose to find the threshold based on the training set to be comparable. In addition, we ran experiments combining activation clipping of the intermediate block output features and our method on a DenseNet-121 with percentile p=90 and showed improvements in average AUROC.

Dice [2]: we followed the proposed protocol for ImageNet suggested in [2], in which we selected the sparsity parameter to be 0.7 for each weight vector of class $y$ of the last linear layer. We compute the energy score on top of the transformed logits. We obtained poor results with the visual transformers and decided not to add results for this method in table 1 as further validation may be needed.

KNN [3]: we followed the recommendations in [3] and set as hyperparameters alpha=1% and k=10 to compute the k-th Euclidean distance w.r.t the training dataset.

**We updated the main table accordingly where our method outperforms SOTA methods by up to 5.4% AUROC.**

**Methods which are not directly comparable to ours**

References [4], [5], [6], among other works cited in the related works, propose specific training to improve OOD detection. [4] relies on models trained in a self-supervised way with a contrastive loss. [5] proposes synthesizing samples from regions of low density during training time as a regularization technique for the learned features. [6] requires classifiers to be attached to the output of each network block. We believe these methods are not directly comparable to this work, **which does not propose any particular custom training setup**. We cite them in the related works section.

**Further improvements to our method**

We acknowledge reviewer 8gLg for making a suggestion that improved our scores significantly. We modified equation 2: instead of taking the predicted score for each class, we compute the weighted sum by the class probabilities. We increase at least 1% in AUROC for every model in the benchmark (see updated Table 1).

\begin{equation}
d_\ell(x; \mu_{\ell})=\sum_{y=1}^{C}\sigma_y(x) \cdot proj_{\mu_{\ell, y}}z_{\ell}
\end{equation}

Where $\sigma_y$ is the softmax function value for class $y$ over the logits $f_\theta(x)$.

**Results on CIFAR-10**

Two out of four reviewers asked for further experiments on a CIFAR benchmark. We ran experiments with a ResNet-18 model trained on CIFAR-10 and evaluated the OOD performance of a few methods compared to ours. We extract features from the residual blocks 2 to 4, the penultimate layer, and logits. The results in AUROC (in percentage) are displayed in the table below.

|              | MSP  | ODIN | Energy | KNN  | ReAct | Ours |
| ------------ | ---- | ---- | ------ | ---- | ----- | ---- |
| CIFAR-100    | 88,0 | 88,8 | 89,1   | 89,9 | 89,7  | 89,4 |
| SVHN         | 91,5 | 91,9 | 92,0   | 94,9 | 94,6  | 99,0 |
| LSUN (c)     | 95,1 | 98,5 | 98,9   | 97,0 | 97,9  | 99,8 |
| LSUN (r)     | 92,2 | 94,9 | 95,3   | 95,8 | 96,7  | 99,8 |
| TinyImageNet | 89,8 | 91,1 | 91,7   | 92,8 | 93,8  | 98,0 |
| Places-365   | 90,1 | 92,9 | 93,2   | 93,7 | 94,7  | 93,6 |
| Textures     | 88,5 | 86,4 | 87,2   | 94,2 | 93,4  | 97,9 |
| Average      | 90,7 | 92,1 | 92,5   | 94,0 | 94,4  | **96,8** |

We added this comparison to the supplementary material.

**References**

[1] Sun, Y., Guo, C. & Li, Y. ReAct: Out-of-distribution Detection With Rectified Activations. In NeurIPS 2021

[2] Sun, Y. & Li, Y. Dice: Leveraging sparsification for out-of-distribution detection. In ECCV 2022

[3] Sun, Y., Ming, Y., Zhu, X. & Li, Y. Out-of-distribution detection with deep nearest neighbors. In ICML 2022

[4] Sehwag, V., Chiang, M., & Mittal, P. SSD: A unified framework for self-supervised outlier detection. In ICLR 2021

[5] Du, X., Wang, Z., Cai, M. & Li, Y. VOS: Learning What You Don't Know by Virtual Outlier Synthesis. In ICLR 2022

[6] Lin, Z., Roy, S. D. & Li, Y. Mood: Multi-level out-of-distribution detection. In CVPR 2021

---

### Decision · Program_Chairs · 2023-01-20

**Decision:**

Reject

**Justification For Why Not Higher Score:**

The current results and improvement lack significance compared to the state-of-the-art, making it questionable whether using intermediate representation and trajectory is beneficial.

**Justification For Why Not Lower Score:**

NA

**Metareview: Summary, Strengths And Weaknesses:**

_Summary_

This paper tackles a challenging and vital problem of out-of-distribution (OOD) detection for a pre-trained multi-layer neural network. The proposed method leverages the input’s representation footprint through a network, and detects OOD samples based on the trajectory similarity w.r.t. the reference trajectory. The reference trajectory is derived from the training population, using the cosine similarity between intermediate representation and layer-wise prototype vector.

_Strengths_

Multiple reviewers found the paper interesting, well-motivated, and clearly written. The technical details are properly documented, which helps with reproducibility. The paper conducted extensive evaluations based on the ImageNet-1k benchmark, and shows strong performance on models trained with cross-entropy loss. The AC agrees that brings a refreshing and new perspective to the research community, which can inspire future works.

_Weaknesses_

During review and discussion, some concerns are brought up too. Given the nature of being feature-based, the proposed methodology has the potential and utility beyond models trained with cross-entropy loss. Reviewers hence suggested the possibility of evaluating the method on models trained with advanced contrastive loss, which can produce more compact representations. The authors actively engaged in the discussion and promptly conducted follow-up evaluations. However, reviewers are concerned about the lack of significance compared to the existing state-of-the-art methods. In particular, the performance is only marginally better than KNN+ (based only on the penultimate layer), which suggests that intermediate features may not add useful information or be utilized optimally. The reviewers eventually were not persuaded due to this major concern.

In summary, it's a beautiful idea and story, but the results in the current status are not sufficiently convincing to justify the merit of the proposed method. The authors are encouraged to explore improved design and target a future conference venue.